# FEDERATED GRANGER CAUSALITY LEARNING FOR INTERDEPENDENT CLIENTS WITH STATE SPACE REPRESENTATION

**Ayush Mohanty**[*][†]**, Nazal Mohamed**[*][†]**,Paritosh Ramanan**[‡]**, & Nagi Gebraeel**[†]

[†]Georgia Institute of Technology, Atlanta, GA, USA; {amohanty42,naz,ngebraeel3}@gatech.edu
[‡]Oklahoma State University, Stillwater, OK, USA; paritosh.ramanan@okstate.edu

## ABSTRACT

Advanced sensors and IoT devices have improved the monitoring and control of complex industrial enterprises. They have also created an interdependent fabric of geographically distributed process operations (clients) across these enterprises. Granger causality is an effective approach to detect and quantify interdependencies by examining how the state of one client affects the states of others over time. Understanding these interdependencies helps capture how localized events, such as faults and disruptions, can propagate throughout the system, potentially leading to widespread operational impacts. However, the large volume and complexity of industrial data present significant challenges in effectively modeling these interdependencies. This paper develops a federated approach to learning Granger causality. We utilize a linear state space system framework that leverages low-dimensional state estimates to analyze interdependencies. This helps address bandwidth limitations and the computational burden commonly associated with centralized data processing. We propose augmenting the client models with the Granger causality information learned by the server through a Machine Learning (ML) function. We examine the co-dependence between the augmented client and server models and reformulate the framework as a standalone ML algorithm providing conditions for its sublinear and linear convergence rates. We also study the convergence of the framework to a centralized oracle model. Moreover, we include a differential privacy analysis to ensure data security while preserving causal insights. Using synthetic data, we conduct comprehensive experiments to demonstrate the robustness of our approach to perturbations in causality, the scalability to the size of communication, number of clients, and the dimensions of raw data. We also evaluate the performance on two real-world industrial control system datasets by reporting the volume of data saved by decentralization.

## 1 INTRODUCTION

The rapid growth of IoT devices and sensor networks has increased the interdependencies between process operations of decentralized systems, such as distributed manufacturing enterprises (Okwudire & Madhyastha (2021), Srai et al. (2020)), supply chains (Lee & Billington (1993), Bernstein & Federgruen (2005)), and power networks(Singh et al. (2018), Kekatos & Giannakis (2013)). These systems comprise geographically distributed assets (e.g., machines and processes) that rely on advanced sensors and IoT technologies for monitoring and control. These technologies often generate large volumes of high-dimensional time-series data that capture the operational state and reliability of various system components. Ensuring reliable system-wide operations is challenging due to **operational interdependencies**, which magnify the effects of fault propagation Bian & Gebraeel (2014) and cascading failures Fu et al. (2023).

This paper focuses on systems with multiple geographically distributed entities— for example, manufacturing and utility plant sites, which we refer to as **clients**—operating in an interconnected manner. We examine the operational interdependencies in these multi-client systems using state-space

---

modeling and causal analysis to better understand their cause-and-effect relationships. Granger causality Granger (1969) is an effective approach to detect and quantify interdependencies by examining how the state of one client affects the states of others over time. This approach captures how localized events, such as faults or disruptions, can propagate throughout the system, potentially leading to widespread operational impacts.

The decentralized nature of data, coupled with its large volume and high dimensionality, presents significant challenges in establishing causality through centralized data analysis. Aggregating data from multiple sources in a central server can become inefficient and impractical as the scale and complexity of the data increase. However, in many applications, it is possible to represent high-dimensional data using low-dimensional state. In the context of causality analysis, low-dimensional state enable the identification of critical interdependencies without aggregating raw data.

In this work, we use linear time-invariant (LTI) state space representation for individual models of a multi-client system. Clients operate independently using only their client-specific information. The measurements (i.e., raw data) at each client are assumed to be high-dimensional. Clients cannot share their measurements, but can only share their low-dimensional state with a central server. Our goal is to develop a federated learning framework that allows a decentralized system of clients to collaboratively learn the off-diagonal blocks of the system's state matrix that represent the cross-client Granger causality—by sharing only their state with a central server. To achieve this, we propose augmenting client models with the off-diagonal information of state matrix through a Machine Learning (ML) based function. To the best of our knowledge, this is the first study on federated granger causality learning. Please refer to Appendix A.1 for preliminaries on *state space modeling*, *Kalman filter*, and *Granger causality*, along their brief mathematical representations.

**Research Objective:** Our objective is to develop a federated learning framework in which the augmented state gradually converges to the centralized state, thus achieving parity between a local and a centralized (oracle) model. Through this process, the decentralized system learns the off-diagonal blocks of the system's state matrix, which capture client interactions by **sharing only their states** with a central server rather than large volumes of high-dimensional measurements.

**Main Contributions:** Our key technical contributions can be delineated as follows:

1. We formulate a federated framework for a multi-client state space system that operates via iterative optimization, where (1) the server learns cross-client Granger causality using low-dimensional states from all clients, and (2) client models, augmented with ML functions, implicitly capture these causality..

2. We prove convergence dependencies between server and client models, and reformulated the server-client iterative framework as a standalone ML algorithm with sublinear and linear convergence rates in its gradient descent.

3. We define a centralized oracle benchmark and proved bounded differences between the ground-truth and learned Granger causality, with matrix bounds under specific conditions.

4. We performed a theoretical analysis to ensure that the communications (both client-to-server, and server-to-client) satisfy differential privacy.

5. Experiments on synthetic data highlight communication efficiency, robustness, and scalability. We also validate the framework on real-world ICS datasets, reporting the volume of data saved by decentralization without compromising the training loss.

## 2 RELATED WORK

Federated learning (FL) is a decentralized machine learning approach where model training occurs across multiple clients, sharing only model updates with a central server. Traditional FL works (McMahan et al. (2017), Yurochkin et al. (2019)) with horizontally partitioned data (Yang et al. (2019b)), where each client has independent data sample but the same feature space. Our approach, however, aligns more with Vertical Federated Learning (VFL), where clients hold different features of the same sample. Since we use time-series data, **in our case the features corresponds to the measurements and the samples refer to the time stamp**. Unlike conventional VFL setting such as (Hu et al. (2019), Gu et al. (2021), Chen et al. (2020), Ma et al. (2023)), Hardy et al. (2017), Yang et al. (2019a), Fang et al. (2021), Wu et al. (2020) which often involves sharing models to

a server and updating the client model, **our framework allows each client to maintain its own model**—based on client-specific observations, without centralizing data or models.

**Split Learning and Multi Task Learning:** Our method shares elements with both (1) split learning (Vepakomma et al. (2018), Poirot et al. (2019), Thapa et al. (2022), Kim et al. (2017)) where different parts of a model are trained separately, and (2) multi-task learning (Smith et al. (2017), Marfoq et al. (2021), and Chen & Zhang (2022)), where tasks share a common representation. However, unlike these methods, **our approach maintains client models' autonomy**.

**Granger Causality:** While Vector Auto Regressive models are widely applied for Granger causality (GC) learning such as Gong et al. (2015), Geiger et al. (2015), Hyvärinen et al. (2010), Huang et al. (2019), Chaudhry et al. (2017), they struggle with systems involving hidden states. State-space (SS) representations offer more flexibility for such systems, but applications of GC in SS models such as Elvira & Chouzenoux (2022), Józsa et al. (2019), Balashankar et al. (2023) are primarily centralized. A comprehensive review of GC can be found in Balashankar et al. (2023). Our framework extends this by enabling federated GC learning in SS systems, where **cross-client causality is inferred by estimating off-diagonal blocks of the state matrix** $A$, assuming client-specific observations through a block diagonal output matrix $C$.

**System Identification :** Traditional system identification literature ( Keesman (2011), Simpkins (2012), Gibson & Ninness (2005)) assumes centralized access to all data, violating the decentralization premise of our framework. Recent methods such as Haber & Verhaegen (2014), Stanković et al. (2015), Mao & He (2022) address this through low-rank and sparse techniques, but still require centralized measurement aggregation or neighbor node knowledge. Our framework bypasses these requirements by **estimating the** $A$ **matrix using low-dimensional states**, retaining the ability to infer causality without moving measurements.

**Distributed Kalman Filter:** Kalman filters estimate latent states from noisy data but face challenges in decentralized settings. Distributed Kalman Filtering such as the ones discussed in Zhang et al. (2022), Xin et al. (2022), Cheng et al. (2021), Olfati-Saber & Shamma (2005), Olfati-Saber (2007), Farina & Carli (2018) allow for decentralized collaboration but typically assumes knowledge of the $A$ matrix or centralization after local filtering. In contrast, our approach **estimates the** $A$ **matrix without prior system knowledge or data movement**.

## 3 PROBLEM SETTING

We assume a server-client framework with $M$ clients having operational interdependencies. Client $m$ observes high dimensional time series measurements $y_m^t \in \mathbb{R}^{D_m}$, utilizes client-specific state matrix $A_{mm}$, and outputs low dimensional states $(h_m^t)_c \in \mathbb{R}^{P_m}$ $(D_m >> P_m)$ via its client model $f_c(.)$, s.t., $(h_m^t)_c = f_c(y_m^t; A_{mm})$. **This model does not capture cross-client causality as it uses only** $A_{mm}$ (and not using $A_{mn} \forall n \neq m$). The framework then proceeds iteratively as follows:

- Client $m$ uses a ML function $f_{ML}(.)$ to augment the client model, producing $(h_m^t)_a$, where $(h_m^t)_a = f_a((h_m^t)_c, f_{ML}(y_m^t; \theta_m))$ and $f_a(.)$ is the augmentation model. **The parameter** $\theta_m$ **encodes cross-client causality**. Client $m$ minimizes the loss $(L_m)_a = \left\| y_m^t - f_c^{-1}((h_m^t)_a) \right\|_2^2$ **w.r.t.** $\theta_m$, then communicates the tuple $[(h_m^t)_a, (h_m^t)_c]$ to the server.

- The server model $f_s(.)$ receives input $(H^t)_c = \left[ (h_1^t)_c, ..., (h_M^t)_c \right]^T$ to produce $(H^t)_s = f_s((H^t)_c; [\hat{A}_{mn}, A_{mm} \forall n \neq m])$. It optimizes the loss $L_s = \left\| (H^t)_a - (H^t)_s \right\|_2^2$ **w.r.t. parameters** $\hat{A}_{mn}$, where $(H^t)_a = \left[ (h_1^t)_a, ..., (h_M^t)_a \right]^T$. **The** $\hat{A}_{mn}$ **are the learned cross-client Granger causality**. The server then communicates the gradient of $L_s$ to the clients.

A discussion on the possible choices of $f_c$, $f_a$, $f_{ML}$ along with the rationale behind our models, is provided in the Appendix A.2. A simplified pictorial description of the aforementioned problem setting is shown in figure 1. A pseudocode for our proposed framework is given in Appendix A.3. Readers can find the code of this paper and associated experiments in `https://github.com/federated-interdependency-learning/fed_granger_causality.git`

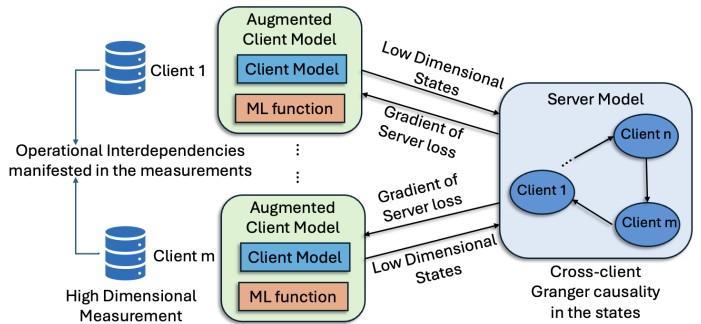

Figure 1: Federated cross-client Granger causality learning framework

# 4 FEDERATED GRANGER CAUSALITY FRAMEWORK

**Nomenclature:** We define $h^t$ as the "***predicted state***," i.e., the state predicted for time $t$ based on measurements $y^{t-1}$, also called the "prior state estimate", "one-step ahead prediction", or "predicted state estimate" in kalman filter literature. The variable $\hat{h}^t$ is the "***estimated state***," based on $y^t$, also known as the "posterior estimate", "updated state estimate", or "current state estimate" in literature.

**Assumption 4.1** (**Client Model**). The client model $f_c(.)$ is a Kalman filter with access to client-specific measurements $y_m \forall m \in \{1, ..., M\}$. It uses **only the diagonal blocks** of the state matrix $A$ and output matrix $C$ (given by $A_{mm}$, and $C_{mm}$ respectively). Equations in the first column of Table 1 define the client model, using only $A_{mm}$, and $C_{mm}$ which are known apriori. If unknown, they can be estimated locally using $y_m$. The estimated and predicted states are $(\hat{h}_m^{t-1})_c$, and $(h_m^t)_c$, with residual $(r_m^t)_c$ and Kalman gain $(K_m)_c$.

• **Insufficiency of Client Models:** The Kalman filter based client models provide optimal state estimation using client-specific measurements. However, they only utilize the diagonal blocks of the state matrix ($A_{mm}$), ignoring the off-diagonal blocks ($A_{mn} \forall n \neq m$). Consequently, the **client models cannot capture the cross-client Granger causality**.

• **Benchmark – A Centralized Oracle:** The *centralized oracle* is a Kalman filter that accesses measurements $y^t$ from all $M$ components. Unlike the client model, **the oracle's state matrix $A$ has non-zero off-diagonal blocks representing cross-client causality**. The third column of Table 1 describe the oracle, where $(\hat{h}^t)_o$ and $(h^t)_o$ are its estimated and predicted states. The matrix $C$ is assumed to be block diagonal. Residual $(r^t)_o$, and Kalman gain $K_o$ are similar to the client model.

Table 1: Equations for the Client Model, Augmented Client Model, and Centralized Oracle

| **Client Model** | **Augmented Client Model** | **Centralized Oracle** |
|---|---|---|
| $(h_m^t)_c = A_{mm} \cdot (\hat{h}_m^{t-1})_c$ | $(h_m^t)_a = A_{mm} \cdot (\hat{h}_m^{t-1})_a$ | $(h^t)_o = A \cdot (\hat{h}^{t-1})_o$ |
| $(r_m^t)_c = y_m^t - C_{mm} \cdot (h_m^t)_c$ | $(r_m^t)_a = y_m^t - C_{mm} \cdot (h_m^t)_a$ | $(r^t)_o = y^t - C \cdot (h^t)_o$ |
| $(\hat{h}_m^t)_c = (h_m^t)_c + (K_m)_c \cdot (r_m^t)_c$ | $(\hat{h}_m^t)_a = (\hat{h}_m^t)_c + \theta_m \cdot y_m^t$ | $(\hat{h}^t)_o = (h^t)_o + K_o \cdot (r^t)_o$ |

## 4.1 AUGMENTED CLIENT MODEL

To address the above insufficieny, we ***augment the client models*** with ML function, enabling learning of Granger causality within their "*augmented states*". The two salient characteristics of this ML function must be as follows: **(1)** the parameter of that function must capture the Granger causality (which is otherwise not captured by the client model), and **(2)** the function must **only utilize the client-specific parameters** $A_{mm}$ and $C_{mm}$, and **client-specific measurements** $y_m$.

We assume a ***additive augmentation*** model s.t., *Augmented Client Model = Client Model + ML function*. Furthermore, as the underlying system is assumed to have a LTI state space representation, we make the following assumption on the ML function to facilitate mathematical insights:

**Assumption 4.2.** The ML function (augmenting the client model) is *linear* in $y_m \ \forall m \in \{1, ..., M\}$

To draw analogies with the client model we state the augmented client model in the second column of Table 1. The estimated and the predicted augmented states are given by $(\hat{h}_m^t)_a$ and $(h_m^t)_a$ respectively. The augmentation is defined in the third equation (of Table 1), where **a linear ML function given by $\theta_m y_m^t$ is added to the estimated state of the client model** to provide the estimated augmented state. $\theta_m$ is the parameter of the ML function.

***Client Loss:*** Similar to the client model, the augmented model also uses client-specific state matrix $(A_{mm})$ and output matrix $(C_{mm})$. The second row of Table 1 defines the augmented client model's residual. The loss function of the augmented client model is given by $(L_m)_a = \|(r_m^t)_a\|_2^2$.

We make the following claim which is validated later in our theoretical analysis and experiments:

**Claim 4.3.** *The client model's parameter $\theta_m$ captures the cross-client Granger causality information of state matrix's off-diagonal blocks $A_{mn} \,\forall n \neq m$, and $m, n \in \{1, ..., M\}$*

At training iteration $k$, the learning of $\theta_m$ uses a gradient descent algorithm as shown in equation 1. There are two partial gradients involved in this step: one corresponding to the augmented client loss $(L_m)_a$ with a learning rate of $\eta_1$, and the other to the server model's loss $L_s$ with a learning rate of $\eta_2$. Effectively, we are **optimizing a weighted sum of $(L_m)_a$ and $L_s$, where the weights are proportional to $\eta_1$ and $\eta_2$.** In equation 1, the second term, $\nabla_{\theta_m^k}(L_m)_a$, can be computed locally at the client. Using the chain rule, we expand the third term of equation 1 to derive equation 2, where $\nabla_{(\hat{h}_m^t)_a} L_s$ is communicated from the server, and $\nabla_{\theta_m^k}(\hat{h}_m^t)_a$ is computed locally at client $m$.

$$\theta_m^{k+1} = \theta_m^k - \eta_1 \cdot \nabla_{\theta_m^k}(L_m)_a - \eta_2 \cdot \nabla_{\theta_m^k} L_s \tag{1}$$

$$= \theta_m^k - \eta_1 \cdot \nabla_{\theta_m^k}(L_m)_a - \eta_2 \cdot \left[\nabla_{(\hat{h}_m^t)_a} L_s \cdot {y_m^t}^\top\right] \tag{2}$$

***Communication from Client to the Server:*** A tuple of the estimated states from the client, and the augmented client model i.e., $[(\hat{h}_m^t)_a, (\hat{h}_m^t)_c]$ are communicated from the client $m$ to the server.

## 4.2 SERVER MODEL

Using the tuple of the estimates states communicated from all $M$ clients, the objective of the server model is to *estimate the state matrix* that encodes cross-client Granger causality in its off-diagonal blocks. We also make the following assumption about the diagonal blocks of that state matrix:

**Assumption 4.4.** The diagonal blocks of the state matrix given by $A_{mm} \in \mathbb{R}^{P_m \times P_m} \,\forall m \in \{1, ..., M\}$ are assumed to be known apriori at the server.

Assumption 4.4 is reasonable as the diagonal blocks are known (or estimated) apriori at the clients, and they need to be *communicated only once* before the onset of the model training.

We use the augmented model's estimated state $(\hat{h}_m^t)_a$, and the diagonal blocks $A_{mm}$ to compute the predicted state $(h_m^t)_a$ (second column of Table 1). These $(h_m^t)_a \,\forall m$ are later **used as training labels** for the server model. On the other hand, a direct consequence of assumption 4.4 is that **only the off-diagonal blocks of the state matrix need to be estimated by the server model**. We denote these estimated off-diagonal blocks are $A_{mn} \in \mathbb{R}^{P_m \times P_n}$, $n \neq m \,\forall m, n \in \{1, ..., M\}$.

***Server Loss:*** The server model inputs states $(\hat{h}_m^t)_c$ (from all $M$ clients), predicts states $(h_m^t)_s$ as output, and compares them against the true labels $(\hat{h}_m^t)_a$. These predictions and labels are concatenated as $(H^t)_s := \left[(h_1^t)_s, ..., (h_M^t)_s\right]^T$, and $(H^t)_a := \left[(h_1^t)_a, ..., (h_M^t)_a\right]^T$ respectively. Then the loss of the server model is given by $L_s = \|(H^t)_a - (H^t)_s\|_2^2$ We now state a claim about the learning of the server model and validate it in later sections:

**Claim 4.5.** *The estimated off-diagonal blocks $\hat{A}_{mn} \,\forall n \neq m$ encode the augmented client model's ML parameter $\theta_m \forall m \in \{1, ..., M\}$*

***Server Model Learning:*** The learning of $A_{mn}$ also uses gradient descent with $\gamma$ as the learning rate. At training iteration $k$, the gradient descent of $A_{mn}$ is given by:

$$\hat{A}_{mn}^{k+1} = \hat{A}_{mn}^k - \gamma \cdot \nabla_{\hat{A}_{mn}^k} L_s \tag{3}$$

***Communication from Server to Client:*** The gradient of the server's loss w.r.t. estimated state of the augmented client model i.e., $\nabla_{(\hat{h}_m^{t-1})_a} L_s \in \mathbb{R}^{P_m}$ is communicated from the server to client $m$.

Table 2: States predicted by the server model and the centralized oracle

| Server Model | Centralized Oracle |
|---|---|
| $(h_m^t)_s = A_{mm}(\hat{h}_m^{t-1})_c + \sum\limits_{n \neq m}^{M} \hat{A}_{mn}(\hat{h}_n^{t-1})_c$ | $(h_m^t)_o = A_{mm}(\hat{h}_m^{t-1})_o + \sum\limits_{n \neq m}^{M} A_{mn}(\hat{h}_n^{t-1})_o$ |

- **Comparison to the Centralized Oracle:** The predicted states of the centralized oracle can be reformulated as shown in the second column of Table 2, which is analogous to the server model. It is important to highlight that, **while the oracle has access to the ground-truth state matrix, the server approximates this matrix** using the states provided by the client model.

## 5 UNDERSTANDING DECENTRALIZATION THROUGH A CENTRALIZED LENS

In this section, we substitute the server model terms with high-dimensional data $y$'s and replace the client model terms with the estimated off-diagonal blocks $\hat{A}_{mn}$'s. This reformulation makes the framework appear "*centralized*" as the $y$'s and $\hat{A}_{mn}$'s are available at one location. However, this is **purely a theoretical tool** for analysis, and in practice, models are trained without any centralization.

**Theorem 5.1** (**Co-dependence**). *At the $(k+1)^{th}$ iteration, the augmented client model's parameter i.e., $\theta_m^{k+1}$ depends on the $k^{th}$ iter. of the server model's parameter i.e., $\hat{A}_{mn}^k, n \neq m$, and vice versa.*

Theorem 5.1 provides the following insights: **(1) the augmented client model encodes the latest estimation of the state matrix during learning**, and **(2) the server model's estimated state matrix depends on the most recent client model.**. Building on these insights and theorem 5.1 we propose corollary 5.2 on the convergence of the ML parameters in both the client and server models.

**Corollary 5.2.** $\theta_m$ *converges if and only if $\hat{A}_{mn}, n \neq m$ converges.*

Next, we state proposition 5.3 that gives the values of the optimal augmented client and server model parameters given by $\theta_m^*$ and $\hat{A}_{mn}^*$, respectively. While the first condition gives the closed-form for $\theta_m^*$ as a function of the knowns; the second condition gives $\hat{A}_{mn}^*$ as a function of $\theta_m^*$.

**Proposition 5.3** (**Optimal model parameters**). *If $C_{mm}A_{mm}$ is of full rank and $y_m^t \neq 0 \forall t$, when augmented client and server model parameter converges to $\theta_m^*$ and $\hat{A}_{mn}^*, n \neq m$, respectively then:*

1. $\mathbb{E}\big[y_m^t - C_{mm}\big(A_{mm}(\hat{h}_m^{t-1})_c + A_{mm}\theta_m^* y_m^{t-1}\big)\big] = 0$

2. $\mathbb{E}\big[\big(A_{mm}\theta_m^* y_m^{t-1}\big)^\top (\hat{h}_n^{t-1})_c - \sum_{p \neq m} \hat{A}_{mp}^* (\hat{h}_p^{t-1})_c^\top (\hat{h}_n^{t-1})_c\big] = 0 \quad \forall n \neq m$

We now offer an alternative perspective, representing the framework as a standalone ML algorithm. Theorem 5.4 unifies the iterative optimization of the server and client models into a unified equation.

**Theorem 5.4** (**Unified framework**). *For any client $m$, the federated framework effectively solves the following recurrent equation:*

$$\Delta^{k+1} = H \cdot \Delta^k + J \tag{4}$$

*where,*

$$\Delta^k := \begin{bmatrix} \text{vec}(\hat{A}_{m1}^k) & \cdots & \text{vec}(\hat{A}_{m(m-1)}^k) & \text{vec}(\hat{A}_{m(m+1)}^k) & \cdots & \text{vec}(\hat{A}_{mM}^k) & \text{vec}(\theta_m^k) \end{bmatrix}^T$$

$$H := \begin{bmatrix} P_{11} & -2\gamma(V_{12}^\top \otimes I) & \cdots & -2\gamma(V_{1M}^\top \otimes I) & \gamma(Q_{m1}^\top \otimes R) \\ \vdots & \vdots & \vdots & \vdots & \vdots \\ -2\gamma(V_{M1}^\top \otimes I) & \cdots & \cdots & P_{MM} & \gamma(Q_{mM}^\top \otimes R) \\ \eta_2(Q_{m1} \otimes R^\top) & \cdots & \cdots & \eta_2(Q_{mM} \otimes R^\top) & (I - G \otimes F) \end{bmatrix}$$

$$J := \begin{bmatrix} 0 & 0 & \cdots & \text{vec}(D) \end{bmatrix}^T \text{ with } D := A_{mm}^T C_{mm}^T (r_m^t)_c y_m^{t-1^T} \text{ and } \otimes \text{ is the Kronecker prod.}$$

$$P_{mm} := (I - 2\gamma(\hat{h}_m^{t-1})_c(\hat{h}_m^{t-1})_c^T) \otimes I \;\; ; \;\; Q_{mn} := y_m^{t-1}(\hat{h}_n^{t-1})_c^T \;\; ; \;\; R := 2A_{mm} \;\; ; \;\; G := y_m^{t-1} y_m^{t-1^T}$$

$$F := \eta_1(2A_{mm}^T A_{mm}) + \eta_2(2A_{mm}^T C_{mm}^T C_{mm} A_{mm}) \;\; ; \;\; V_{mn} := (\hat{h}_n^{t-1})_c(\hat{h}_m^{t-1})_c^T$$

The augmented client's and server's loss functions i.e., $(L_m)_a$ and $L_s$ are convex in $\theta_m$ and $\hat{A}_{mn}$, respectively, so their stationary points are global minima. Since $\theta_m$ and $\hat{A}_{mn}$ are elements of $\Delta$ in the recurrence equation 4, the stationary values can also be derived from its asymptotic behavior. Lemma 5.5 provides the asymptotic convergence condition of equation 4 and its stationary values.

**Lemma 5.5** (**Convergence of framework**). *The federated framework converges if and only if $\rho(H) < 1$. Furthermore, the stationary value of $\Delta$ is given by $\Delta^* = (I - H)^{-1} J$*

Upon algebraic manipulation of equation 4 we can obtain the following recurrent linear equation:

$$\Delta^{k+1} = \Delta^k - (I - H) \cdot [\Delta^k - ((I - H)^{-1} J)] = \Delta^k - (I - H) \cdot [\Delta^k - \Delta^*] \quad (5)$$

Equation 5 is **analogous to gradient descent of the proposed federated framework** parameterized by $\Delta$. Let $L_f$ represent the loss function of the federated framework, whose **explicit functional form is unknown**. Under special conditions on $L_f$ we analyze the convergence rate of the gradient descent in the joint space of $\hat{A}_{mn}$ and $\theta_m$. Leveraging well established results on gradient descent we provide theorems 5.6 and 5.7 to discuss conditions for sub linear and linear convergence.

**Theorem 5.6** (**Sub linear conv.**). *If $L_f$ is convex, and $\mathcal{L}$-Lipschitz smooth in the joint space of $\hat{A}_{mn}$, and $\theta_m$, with $H$ chosen s.t., $\|I - H\| \leq 1$, then convergence rate of $L_f$ is $O(1/k)$*

**Theorem 5.7** (**Linear conv.**). *If $L_f$ is $\mathcal{L}$-Lipschitz smooth, and $\mu$-strongly convex in the joint space of $\hat{A}_{mn}$, and $\theta_m$ with $H$ chosen s.t., $\|I - H\| \leq \frac{2\mathcal{L}}{\mu + \mathcal{L}}$, then convergence rate of $L_f$ is $O((1 - \frac{\mu}{\mathcal{L}})^k)$*

# 6 ASYMPTOTIC CONVERGENCE TO THE CENTRALIZED ORACLE

We assume that **the centralized oracle is convergent** i.e., it has a zero steady state error. We first analyze the convergence of the states learned using our approach to the oracle. Theorem 6.1 shows that the predicted states of the augmented client model converge in expectation to the oracle.

**Theorem 6.1.** *Let $(\hat{h}_m^{t,k})_a := (\hat{h}_m^t)_c + \theta_m^k y_m^t$, and $(h_m^{t,k})_a := A_{mm} \cdot (\hat{h}_m^{t,k})_a$ (see table 1). Then, for a full rank $C_{mm}$ we have the convergence: $\lim_{k \to \infty} \mathbb{E}[(h_m^{t,k})_a - (h_m^t)_o] = 0 \ \forall m \in \{1, ..., M\}$*

Proposition 6.2 shows that the norm difference between the estimated states of centralized oracle and client model is bounded in expectation. We use this bound to establish the subsequent results.

**Proposition 6.2.** *If the client model satisfies $\rho(A_{mm} - A_{mm}(K_m)_c C_{mm}) < 1$ then $\exists \delta_{max}^m$ such that the following bound holds: $\mathbb{E}[\|(\hat{h}_m^t)_o - (\hat{h}_m^t)_c\|] \leq \delta_{max}^m \forall m \in \{1, ..., M\}$*

Next, we analyze the error in estimating the state matrix. For any two clients $m$ and $n$ with $n \neq m$, let $\hat{A}_{mn}^*$ be the stationary point for the off-diagonal block of the estimated state matrix. Let $A_{mn}$ be the ground truth for those off-diagonal blocks. Then theorem 6.3 and corollary 6.4 provide upper bound on the estimation error of the state matrix **without apriori knowledge of its ground-truth**

**Theorem 6.3.** *If $\rho(A_{mm} - A_{mm}(K_m)_c C_{mm}) < 1$ then, $\mathbb{E}\left[\|\sum_{n \neq m}[\hat{A}_{mn}^* - A_{mn}] \cdot (\hat{h}_n^{t-1})_o\|\right] \leq$*
*$\|A_{mm}\delta_{max}^m\| + \|\sum_{n, n \neq m} \hat{A}_{mn}^* \delta_{max}^n\| \ \forall m \in \{1, ..., M\}$*

**Corollary 6.4.** *If $\exists \sigma_{min}^n$ s.t., $\sigma_{min}^n := \min_{n \neq m} \mathbb{E}[\|(\hat{h}_n^{t-1})_o\|]$ and the vectors $[\hat{A}_{mn}^* - A_{mn}] \cdot (\hat{h}_n^{t-1})_o$ are collinear $\forall n \in \{1, ..., M\}$ and $n \neq m$ then $\forall m \in \{1, ..., M\}$,*
*$\|\sum_{n \neq m}[\hat{A}_{mn}^* - A_{mn}]\|_F \leq \frac{1}{\sigma_{min}^n} \cdot \left(\|A_{mm}\delta_{max}^m\| + \|\sum_{n, n \neq m} \hat{A}_{mn}^* \delta_{max}^n\|\right)$*

# 7 PRIVACY ANALYSIS

In this section, we establish two theoretical results to ensure differential privacy of our framework.

**Client:** Each client $m$ independently perturbs its client model's, and augmented client model's estimated states before sending them to the server as follows:

$$\tilde{h}_{m,c}^t = (\hat{h}_m^t)_c + \mathcal{N}(0, \sigma_c^2 I), \quad \tilde{h}_{m,a}^t = (\hat{h}_m^t)_a + \mathcal{N}(0, \sigma_a^2 I),$$

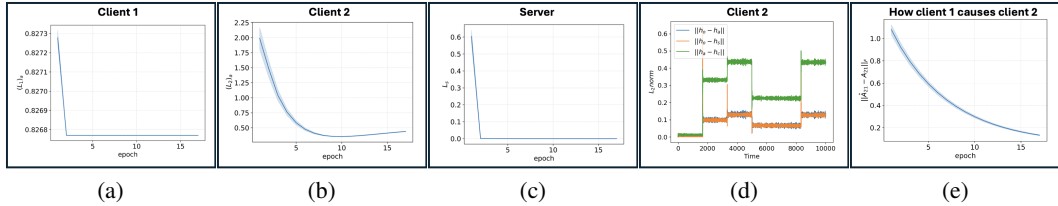

Figure 2: Loss functions at (a) client 1, (b) client 2, and (c) server during the first mean-shift. (e) $l_2$ norm diff. between states of centralized oracle, server, client, augmented client models of client 2 (d) and evolution of Frobenius norm difference between estimation and ground-truth value of $A_{21}$

Table 3: Cross-client Granger causality – estimated ($\hat{A}$) vs ground truth ($A$)

| How does client 1 Granger causes client 2? | | How does client 2 Granger causes client 1? | |
|---|---|---|---|
| Estimated $\hat{A}_{21}$ | Ground truth $A_{21}$ | Estimated $\hat{A}_{12}$ | Ground truth $A_{12}$ |
| $\begin{bmatrix} 0.2793 & 0.2441 \\ 0.3351 & 0.3298 \end{bmatrix}$ | $\begin{bmatrix} 0.25 & 0.25 \\ 0.25 & 0.25 \end{bmatrix}$ | $\begin{bmatrix} 0.0186 & -0.0113 \\ 0.0102 & 0.0010 \end{bmatrix}$ | $\begin{bmatrix} 0 & 0 \\ 0 & 0 \end{bmatrix}$ |

**Server:** The server computes the gradient $\nabla_{(\hat{h}_m^t)_a} L_s$ and applies gradient clipping with clipping threshold $C_g$ and Gaussian noise addition as follows: $\tilde{g}_m^t = \text{Clip}\left(\nabla_{(\hat{h}_m^t)_a} L_s, C_g\right) + \mathcal{N}(0, \sigma_g^2 I)$,

**Theorem 7.1 (Client-to-Server Comm.).** *At each time step $t$, the mechanisms by which client $m$ sends $\tilde{h}_{m,c}^t$ and $\hat{h}_{m,a}^t$ to the server satisfy $(\varepsilon, \delta)$-differential privacy with respect to $y_m^t$, provided that the noise standard deviations satisfy the following with $\varepsilon = \varepsilon_c + \varepsilon_a$ and $\delta = \delta_c + \delta_a$:*

$$\sigma_c \geq \frac{2 B_y B_K \sqrt{2 \ln(1.25/\delta_c)}}{\varepsilon_c}, \qquad \sigma_a \geq \frac{2 B_y (B_K + B_\theta) \sqrt{2 \ln(1.25/\delta_a)}}{\varepsilon_a},$$

**Theorem 7.2 (Server-to-Client Comm.).** *At each time step $t$, the mechanism by which the server sends $\tilde{g}_m^t$ to client $m$ satisfies $(\varepsilon, \delta)$-differential privacy with respect to any single client's data (states), provided that the noise standard deviation satisfies: $\sigma_g \geq \frac{2 C_g \sqrt{2 \ln(1.25/\delta)}}{\varepsilon}$.*

Please refer to Appendix A.5 for a discussion on privacy, along with the meaning of $B_y$, $B_K$, $B_\theta$.

## 8 EXPERIMENTS: SYNTHETIC DATASET

**Dataset Description & Experimental Settings:** The synthetic data simulates a multi-client linear state space system with "mean-shifts" representing an anomaly or change in operating condition. The absence of off-diagonal blocks of $A$ matrix in client model affects the states only after a mean-shift. This can be visualized in figure 2(d) whose details are explained later in this section. We use the same client and server models discussed in section 4. All models are regularized to ensure feasible solutions. Experiments began by checking convergence stability (ensuring $\rho(H) < 1$), adjusting hyperparameters if needed. Unless noted otherwise, experiments used two clients ($M = 2$) with $D_m = D = 8$, $P_m = P = 2 \, \forall m$. Exceptions apply to scalability studies.

**Learning Granger Causality:** We train the framework for a two-client system where ***the states of client 1 Granger-causes client 2 and not vice versa i.e, $A_{21}^{train} \neq 0$ and $A_{12}^{train} = 0$***. The training losses are given in figure 2(a)-(c). The $l_2$ norm differences between **(1)** the client model and the augmented client model, **(2)** the centralized oracle and the augmented client model, and **(3)** the centralized oracle and the server model are shown in figures 2(d). These plots validate claims 4.3, 4.5, and theorem 6.1. Figure 2(e) track the Frobenius norm difference between the ground-truth and estimated state matrices, which decreases during training, further validating theorem 6.3 and corollary 6.4. The estimated and ground truth $A$ matrices are mentioned in Table 3.

**Robustness to Perturbation in Causality:** We introduce perturbations to all elements of the off-diagonal blocks of $A$ matrix to assess the framework's robustness. Specifically all elements of

Table 4: Robustness to perturbations in causality and change in network topology

| Perturbation $\Rightarrow$ | $\epsilon = 5\%$ | | $\epsilon = 45\%$ | | $\epsilon = 85\%$ | | $\epsilon = 125\%$ | |
|---|---|---|---|---|---|---|---|---|
| **Framework** | $(L_2)_a$ | $L_s$ | $(L_2)_a$ | $L_s$ | $(L_2)_a$ | $L_s$ | $(L_2)_a$ | $L_s$ |
| No client aug. | – | $10^{-5}$ | – | $10^{-5}$ | – | $10^{-5}$ | – | $10^{-5}$ |
| No server model | 0.22 | – | 0.58 | – | 0.88 | – | 1.135 | – |
| Pre-trained client | 0.22 | 0.007 | 0.58 | 0.015 | 0.88 | 0.022 | 1.135 | 0.028 |
| **Our method** | 0.39 | 0.003 | 0.57 | 0.007 | 0.76 | 0.010 | 0.93 | .013 |
| Net. Topology $\Rightarrow$ | Preserving | | Reversing | | Eliminating | | Bidirectional | |
| **Framework** | $(L_2)_a$ | $L_s$ | $(L_2)_a$ | $L_s$ | $(L_2)_a$ | $L_s$ | $(L_2)_a$ | $L_s$ |
| No client aug. | – | $10^{-5}$ | – | $10^{-5}$ | – | $10^{-5}$ | – | $10^{-5}$ |
| No server model | 0.182 | – | 0.24 | – | 0.279 | – | 0.127 | – |
| Pre-trained client | 0.182 | 0.006 | 0.24 | 0.065 | 0.279 | 0.012 | 0.127 | 0.014 |
| **Our method** | 0.37 | 0.003 | 0.35 | 0.033 | 0.40 | 0.006 | 0.34 | 0.008 |

training block matrix were perturbed to generate test data s.t., $[A_{21}^{test}]_i = [A_{21}^{train}]_i + \epsilon_i \;\; \forall i \in 1, ..., P$ where, $\epsilon_i = \{5, 45, 85, 125\}\%$ of $[A_{21}^{train}]_i \;\; \forall i$.

**Robustness to Change in Network Topology:** We trained the system with $(A_{21}^{(train)} \neq 0,$ $A_{12}^{(train)} = 0)$. We now modify the test data topology under four conditions: **(1)** underline{preserving} causality $(A_{21}^{(test)} = A_{21}^{(train)}, A_{12}^{(test)} = 0)$, **(2)** reversing causality $(A_{21}^{(test)} = 0, A_{12}^{(test)} \neq 0)$, **(3)** eliminating causality $(A_{21}^{(test)} = 0, A_{12}^{(test)} = 0)$, **(4)** using bidirectional causality $(A_{21}^{(test)} \neq 0, A_{12}^{(test)} \neq 0)$.

**Interpreting Robustness Results:** When $A$ matrix changes during testing, we expect $L_s$ and $(L_2)_a$ to be higher (than training). While a high $L_s$ refers to a flag by the server model, a high $(L_2)_a$ refers to a flag by the client (client 2's) model. We say a framework has learned causality if both server and client models flag with alteration in causality (i.e., either perturbation or change in topology).

The testing losses for the robustness studies are shown in Table 4. For "our method", both $(L_2)_a$ and $L_s$ increase with increase in $\epsilon$, further validating the claims 4.3 and 4.5. Table 4 also shows that with change in the topology, $L_s$ increases for "our method". The reverse causality shows the highest $L_s$ values, thereby inferring that the server model learns causality and flags with alterations in causality. Furthermore, the client model does not show a clear trend to change in network topology. Thus it can generate false alarms (that there is change in causality) if inferencing is done only based on $(L_2)_a$. Further investigation is need to analyze the reasons behind this observation.

**Baselines:** We benchmark our framework against three other versions of our framework: **(1)** same framework without the client augmentation (this underscores the limitations of ignoring the effects of interdependencies with other clients), **(2)** same framework but without the server model (this highlights the importance of server model in improving the client augmentation), **(3)** pre-trained client models as discussed in Ma et al. (2023) (this demonstrates the importance of the iterative optimization in estimating the true interdependencies). Given the constraints on space, we present the interpretation of our framework's performance relative to these baselines in Appendix A.8.

**Scalability Studies:** We increase the dimensions of the measurements $D_m$, keeping the state dimensions $P_m = 2$, and $M = 2$. We trained and tested our framework with $D = \{2, 4, 8, 16, 32\}$. We also validated the scalability w.r.t. the number of clients, by scaling $M$ to $\{2, 4, 8, 16, 32\}$ by fixing $D_m = D = 8$ and $P_m = P = 2 \;\; \forall m$. The results for both studies are reported in Table 5. While there is a trend observed for scalability w.r.t. $D$, none of the studies shows any drastic increase in the order of magnitude for $L_s$, thereby demonstrating that the framework is scalable in both measurement dimension and number of clients.

Table 5: Server loss $L_s$ by scaling measurement dim. $D$ and number of clients $M$

| **Measurement Dim. ($D$)** | | | | **No. of Clients ($M$)** | | | |
|---|---|---|---|---|---|---|---|
| $D = 16$ | $D = 32$ | $D = 64$ | $D = 128$ | $M = 2$ | $M = 4$ | $M = 8$ | $M = 16$ |
| 0.0027 | 0.0061 | 0.0090 | 0.0084 | 0.0003 | 0.0001 | 0.0026 | 0.0004 |

Table 6: Description of the real-world industrial control system (ICS) datasets

| Dataset | $M$ | $\sum_{m=1}^{M} D_m$ | Dataset Description |
|---------|-----|---------------------|---------------------|
| HAI | 4 | 86 | Steam turbine-power & pumped-storage hydropower generation |
| SWaT | 6 | 51 | Water treatment facility |

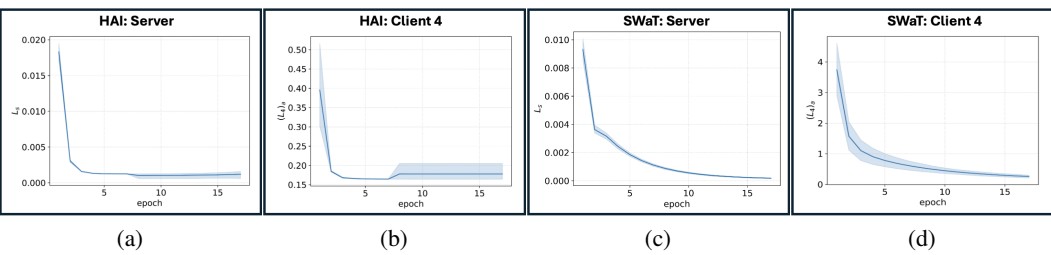

(a)     (b)     (c)     (d)

Figure 3: (a) Server loss and (b) Client 4's loss for HAI dataset, and (c) Sever loss and (d) Client 4's loss for SWaT dataset at a randomly chosen time

# 9 EXPERIMENTS: REAL WORLD DATASETS

**Datasets:** We utilized two ICS datasets – **(1)** HAI: Hardware-the-loop Augmented Industrial control system Shin et al. (2023), and **(2)** SWaT: Secure Water Treatment Mathur & Tippenhauer (2016). For both of the datasets, clients in our framework corresponds to the processes in the datasets. Details of the raw data are given in Table 6.

**Preprocessing:** We first select the measurements with a high ($\geq 0.3$) pairwise Pearson correlation with measurements from other clients. For each client $m$ in the real-world dataset, client model was obtained as follows: **(1)** Apply SVD to the measurement $y_m$ and select the top $P$ right singular vectors as the low-dimensional states, **(2)** Store $C_{mm}$ as the product of the left singular vectors and singular values up to $P$ dimensions, **(3)** Fit a VAR model of the low-dimensional states to compute $A_{mm}$. The framework was then trained on nominal data, free of attacks.

**Granger Causality Learning:** We used the same state dimension $P$ for all $M$ clients in either dataset. The server loss and the augmented client loss (at a randomly chosen client) during training are provided in figure 3. We do not have a ground truth $A$ matrix for any of the real-world datasets. We first perform a centralized estimation of $A$ matrix and considered that as our ground-truth. For $P = 2$, the estimation error between federated and centralized method is provided in Table 7. We also report the amount of data volume saved (in bytes) by utilizing our federated learning approach.

Table 7: Comparison federated and centralized method for real-world datasets

| Dataset | $\|\hat{A} - A\|_F$ | Data saved per comm. round |
|---------|---------------------|----------------------------|
| HAI | 0.8140 | 144 bytes |
| SWaT | 3.0816 | 176 bytes |

# 10 CONCLUSION AND LIMITATIONS

This paper introduces a federated framework for learning Granger causality in distributed systems, addressing high-dimensional data challenges. Using a linear state-space representation, cross-client Granger causality is modeled as off-diagonal terms in the state matrix. The framework augments client models with server-derived causal insights, improving accuracy. We provide theoretical guarantees, demonstrate convergence rates, and include a differential privacy analysis to ensure data security. Experiments on synthetic and real-world datasets validate the framework's robustness and scalability. Limitations and potential future extensions are discussed in Appendix A.9.

ACKNOWLEDGMENTS

This effort is supported by NASA under grant number 80NSSC19K1052 as part of the NASA Space Technology Research Institute (STRI) Habitats Optimized for Missions of Exploration (HOME) 'SmartHab' Project. Any opinions, findings, and conclusions or recommendations expressed in this material are those of the authors and do not necessarily reflect the views of the National Aeronautics and Space Administration.

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

# A  APPENDIX

## CONTENTS

## A.1 PRELIMINARIES

• **State-Space Model:** A state-space model is a mathematical framework used to represent the dynamics of a physical system as a set of measurements $\mathbf{y}^t$, states $\mathbf{h}^t$, all related through a difference equation. It characterizes the system's evolution over time by describing how its state transitions and generates measurements, using two primary equations: the state transition equation (6) and the observation (measurement) equation (7).

$$\mathbf{h}^t = A\mathbf{h}^{t-1} + \mathbf{w} \tag{6}$$

$$\mathbf{y}^t = C\mathbf{h}^t + \mathbf{v} \tag{7}$$

Here, $\mathbf{w}$ and $\mathbf{v}$ are the i.i.d. Gaussian noise added to the states and measurements respectively. The matrices $A$ and $C$ are called the state transition and observation (measurement) matrices, respectively. The above representation is a linear time-invariant (LTI) model, as **(1)** the equations are linear, **(2)** $A$, $C$, distributions of $\mathbf{w}$ and $\mathbf{v}$ are assumed to be stationary.

In a system represented by state-space model, knowledge of the state $\mathbf{h}^t$ is fundamental for understanding the system dynamics. However in real world applications, one often observes only the measurements $\mathbf{y}^t$ without direct access to the states $\mathbf{h}^t$. Estimation of $\mathbf{h}^t$ using $\mathbf{y}^t$ is typically done by a Kalman filter, explained next.

• **Kalman Filter:** A Kalman filter (KF) is an algorithm designed to provide optimal linear estimates of the states based on measurement $\mathbf{y}^t$. A brief walk-through of the steps involved in a KF is explained in the next paragraph.

$$h^t = A \cdot \hat{h}^{t-1} \tag{8}$$

$$r^t = y^t - C \cdot h^t \tag{9}$$

$$\hat{h}^t = h^t + K \cdot r^t \tag{10}$$

At time $t$, the KF predicts the state $h^t$ using the previous estimate $\hat{h}^{t-1}$ (eq. 8). Upon receiving a new measurement $y^t$, it computes the residual $r^t$, the difference between $y^t$ and the predicted measurement $C \cdot h^t$ (eq. 9). The state estimate $\hat{h}^t$ is then updated by adding a correction term $K \cdot r^t$ to its predicted state $h^t$ (eq. 10), with Kalman gain $K$ determining the weight of the residual.

KF operates under the **assumption of complete knowledge of $A$ matrix**, and other parameters such as $C$, noise covariances, etc. Without a given $A$ matrix, implementation of KF mandates its prior estimation. Granger causality is one of the techniques used to explicitly estimate the $A$ matrix.

• **Granger Causality:** A time series $\mathbf{h}_1$ is said to "*granger cause*" another time series $\mathbf{h}_2$, if $\mathbf{h}_1$ can predict $\mathbf{h}_2$. In the context of state space models, the **state matrix $A$ characterizes the granger causality**. For example, in a two state sytem, given in eq.(11), state $\mathbf{h}_2$ is said to "*granger cause*" state $\mathbf{h}_1$ if $A_{12} \neq 0$

$$\begin{bmatrix} \mathbf{h}_1^t \\ \mathbf{h}_2^t \end{bmatrix} = \begin{bmatrix} A_{11} & A_{12} \\ A_{21} & A_{22} \end{bmatrix} \cdot \begin{bmatrix} \mathbf{h}_1^{t-1} \\ \mathbf{h}_2^{t-1} \end{bmatrix} + \begin{bmatrix} \mathbf{w}_1 \\ \mathbf{w}_2 \end{bmatrix} \tag{11}$$

Learning granger causality in a state space model involves estimation of $A$ matrix. Estimating $A$ matrix often involves centralizing the measurements $\mathbf{y}^t$ and performing a maximum-likelihood estimation. However, in systems with decentralized components (clients), centralizing the measurements $\mathbf{y}^t$ from each component can be a challenge. This is especially true when $\mathbf{y}^t$ is high dimensional.

Decentralized learning of $A$ matrix is the primary objective of our paper.

## A.2 DEFINING FUNCTIONS $f_c(.), f_a(.), f_{ML}(.)$

1. **Client Model $f_c(.)$:** The client model can be any machine learning model, such as neural networks, linear regression, or a state space model. For mathematical tractability, we use a state-space model.

   One example of the client model can be an anomaly detection model. In this paper, we assume a client model $f_c(.)$ is **trained independently using only client-specific measurements**.

2. **ML Function** $f_{ML}(.)$**:**

   The ML function $f_{ML}(.)$ is a machine learning model that specifically captures the effects of interactions (granger causality) with other clients. It enhances the awareness of the local client model towards interdependencies with other clients.

   In our implementation, we use linear regression as $f_{ML}$. At client $m$, $f_{ML}$ takes only the client-specific measurements $y_m^t$ as input but encodes information from other clients during the gradient update process. This allows the model to benefit from collective insights while preserving data privacy.

3. **Augmented Client Model** $f_a(.)$**:**

   The augmented client model $f_a(.)$ **combines the client model** $f_c$ **and the ML function** $f_{ML}$ **to enhance the client model**. In our case, this augmentation is achieved through a simple addition:

   $$f_a(.) = f_c(.) + f_{ML}(.)$$

   However, more complex models like neural networks or higher-order polynomials can also be used for augmentation. The output of $f_a(.)$ is designed to have the same dimension as $f_c(.)$, ensuring that it serves as a direct enhancement to the client model without altering its fundamental structure.

### A.3   PSEUDOCODE

The pseudocode for running the client model is given in algorithm 1. The client model is run independent at each client without any federated approach. The output of the client model i.e., the estimated states $(\hat{h}_m^t)_c \forall m, t$ are utilized in the federated Granger causal learning, whose pseducodoe is Ts given in algorithm 2. $T$ is the time series length, $epoch$ is the maximum training epochs, $tol$ is the stoppage tolerance for the server loss and $k$ is the iteration index.

---

**Algorithm 1** Client Model

---

1: **Inputs:**$T$, $A_{mm}$, $C_{mm}$
2: **Choose at Client** $m$: $(K_m)_c$
3:    **for** $t = 1$ to $T$ **do**
4:       $(h_m^t)_c \leftarrow A_{mm} \cdot (\hat{h}_m^{t-1})_c$
5:       $(r_m^t)_c \leftarrow y_m^t - C_{mm} \cdot (\hat{h}_m^t)_c$
6:       $(\hat{h}_m^t)_c \leftarrow (h_m^t)_c + (K_m)_c \cdot (r_m^t)_c$
7:    **end for**

---

### A.4   PROOFS

#### A.4.1   PROOF OF THEOREM 5.1:

We know that the server loss $L_s$ is given by,

$$L_s = \|H_a^t - H_s^t\|_2^2 \tag{12}$$

We know from the definition of augmented client states that,

$$H_a^t = \begin{pmatrix} (h_1^t)_a \\ . \\ . \\ . \\ (h_M^t)_a \end{pmatrix} = \begin{pmatrix} A_{11}(\hat{h}_1^{t-1})_a \\ . \\ . \\ . \\ A_{MM}(\hat{h}_M^{t-1})_a \end{pmatrix} \tag{13}$$

We also know from the definition of server states $H_s$ that,

$$H_s^t = \begin{pmatrix} (h_1^t)_s \\ . \\ . \\ . \\ (h_M^t)_s \end{pmatrix} = \begin{pmatrix} A_{11}(\hat{h}_1^{t-1})_c + \sum_{p\neq 1} \hat{A}_{1p}^k(\hat{h}_p^{t-1})_c \\ . \\ . \\ . \\ A_{MM}(\hat{h}_M^{t-1})_c + \sum_{p\neq M} \hat{A}_{Mp}^k(\hat{h}_p^{t-1})_c \end{pmatrix} \tag{14}$$

---

**Algorithm 2** Federated Learning of Granger Causality

---

1: **Inputs**: $T, A_{mm}, C_{mm}, (\hat{h}_m^t)_c \forall m \in \{1, ..., M\}, t \in \{1, ..., T\}$
2: **Choose**: *epoch*, *tol*, $k = 0$
3: **Initialize at Server**: $\{A_{mn}^0\}_{m \neq n}$
4: **Choose** at Server: $\gamma$
5: **Initialize at Client** $m$: $\theta_m^0$
6: **Choose** at Client: $\eta_1, \eta_2$
7: **while** $k < epoch \cdot T$ or $L_s > tol$ **do**
8:     **for** $t = 1$ to $T$ **do**

9:         **for** each client $m$ **do**
10:             $(h_m^t)_a \leftarrow A_{mm} \cdot (\hat{h}_m^{t-1})_a$
11:             $(\hat{h}_m^t)_a \leftarrow (\hat{h}_m^t)_c + \theta_m^k \cdot y_m^t$
12:             Send $[(\hat{h}_m^t)_a, (\hat{h}_m^t)_c]$ to the server
13:         **end for**

14:         **At** the server:
            $(H^t)_a \leftarrow [(h_1^t)_a, ..., (h_M^t)_a]^T$ and $(H^t)_c \leftarrow [(h_1^t)_c, ..., (h_M^t)_c]^T$
            $(H^t)_s$ is computed using $(h_m^t)_s = A_{mm}(\hat{h}_m^{t-1})_c + \sum\limits_{n \neq m}^{M} \hat{A}_{mn}^k (\hat{h}_n^{t-1})_c$

            $L_s \leftarrow \|(H^t)_a - (H^t)_s\|_2^2$
            Send $\nabla_{(\hat{h}_m^{t-1})_a} L_s$ to the client $m$
            $\hat{A}_{mn}^{k+1} \leftarrow \hat{A}_{mn}^k - \gamma \cdot \nabla_{\hat{A}_{mn}^k} L_s$

15:         **for** each client $m$ **do**
16:             $\theta_m^{k+1} \leftarrow \theta_m^k - \eta_1 \cdot \nabla_{\theta_m^k}(L_m)_a - \eta_2 \cdot \nabla_{\theta_m^k} L_s$
17:         **end for**

18:     **end for**
19: **end while**

---

Therefore the server loss $L_s$ is given by,

$$L_s = \left\| \begin{pmatrix} A_{11}(\hat{h}_1^{t-1})_a - [A_{11}(\hat{h}_1^{t-1})_c + \sum_{n \neq 1} \hat{A}_{1n}^k (\hat{h}_n^{t-1})_c] \\ . \\ . \\ . \\ A_{MM}(\hat{h}_M^{t-1})_a - [A_{MM}(\hat{h}_M^{t-1})_c + \sum_{n \neq M} \hat{A}_{Mn}^k (\hat{h}_n^{t-1})_c] \end{pmatrix} \right\|_2^2 \tag{15}$$

• **(1) Update of Server Model Parameter:**

We take derivative of $L_s$ w.r.t. $\hat{A}_{mn}^k$ to obtain,

$$\nabla_{\hat{A}_{mn}^k} L_s = -2 \left( A_{mm}(\hat{h}_m^{t-1})_a - [A_{mm}(\hat{h}_m^{t-1})_c + \sum_{n \neq m} \hat{A}_{mn}^k (\hat{h}_n^{t-1})_c] \right) (\hat{h}_n^{t-1})_c^T \tag{16}$$

Substituting equation 16 in equation 3 (i.e., gradient descent of $\hat{A}_{mn}$) we obtain,

$$\hat{A}_{mn}^{k+1} = \hat{A}_{mn}^k + 2\gamma \left( A_{mm}(\hat{h}_m^{t-1})_a - [A_{mm}(\hat{h}_m^{t-1})_c + \sum_{p \neq m} \hat{A}_{mp}^k (\hat{h}_p^{t-1})_c] \right) (\hat{h}_n^{t-1})_c^T \tag{17}$$

We know from the definition of augmented client model that,

$$(\hat{h}_m^{t-1})_a = (\hat{h}_m^{t-1})_c + \theta_m^k y_m^{t-1} \tag{18}$$

Substituting equation 18 in equation 17 we obtain,

$$\hat{A}_{mn}^{k+1} = \hat{A}_{mn}^k + 2\gamma [A_{mm}\theta_m^k y_m^{t-1} - \sum_{n \neq m} \hat{A}_{mn}^k (\hat{h}_n^{t-1})_c] (\hat{h}_n^{t-1})_c^T \tag{19}$$

From equation 19 we observe that the $(k + 1)^{th}$ iteration of server model parameter i.e., $\hat{A}_{mn}^{k+1}$ is dependent on the $k^{th}$ iteration of augmented client model parameter i.e., $\theta_m^k$.

• (2) **Update of Augmented Client Model Parameter:**

At client $m$, the client loss $(L_m)_a$ is given by,

$$(L_m)_a = \left\| y_m^t - C_{mm} \cdot A_{mm} \cdot \left( (\hat{h}_m^{t-1})_c + \theta_m^k y_m^{t-1} \right) \right\|_2^2 \tag{20}$$

The analytical derivative of $(L_m)_a$ w.r.t. $\theta_m^k$ is given by,

$$\nabla_{\theta_m^k} (L_m^t)_a = -2 \cdot \left( C_{mm} \cdot A_{mm} \right)^T \cdot \left( y_m^t - C_{mm} \cdot A_{mm} \cdot \left[ (\hat{h}_m^{t-1})_c + \theta_m^k y_m^{t-1} \right] \right) \cdot y_m^{t-1^T} \tag{21}$$

Computing derivative of $L_s$ w.r.t. $(\hat{h}_m^{t-1})_a$ we have,

$$\nabla_{(\hat{h}_m^{t-1})_a} L_s = 2 \cdot A_{mm}^T \cdot \left( A_{mm} \left[ (\hat{h}_m^{t-1})_a - (\hat{h}_m^{t-1})_c \right] - \sum_{n \neq m} \hat{A}_{mn}^k (\hat{h}_n^{t-1})_c \right) \tag{22}$$

The derivative of $(\hat{h}_m^{t-1})_a$ w.r.t. $\theta_m^k$ is given by,

$$\nabla_{\theta_m^k} (\hat{h}_m^{t-1})_a = y_m^{t-1^T} \tag{23}$$

Substituting equations 21, 22, and 23 in equation 2 (i.e., gradient descent of $\theta_m^k$) we obtain,

$$\theta_m^{k+1} = \theta_m^k + 2 \cdot \eta_1 \cdot \left( C_{mm} \cdot A_{mm} \right)^T \cdot \left( y_m^t - C_{mm} \cdot A_{mm} \cdot \left[ (\hat{h}_m^{t-1})_c + \theta_m^k y_m^{t-1} \right] \right) \cdot y_m^{t-1^T}$$
$$- 2 \cdot \eta_2 \cdot A_{mm}^T \cdot \left( A_{mm} \left[ (\hat{h}_m^{t-1})_a - (\hat{h}_m^{t-1})_c \right] - \sum_{n \neq m} \hat{A}_{mn}^k (\hat{h}_n^{t-1})_c \right) \cdot y_m^{t-1^T} \tag{24}$$

From equation 24 we observe that the $(k + 1)^{th}$ iteration of augmented client model parameter i.e., $\theta_m^k$ is dependent on the $k^{th}$ iteration of server model parameter i.e., $\hat{A}_{mn}^k$.

### A.4.2 PROOF OF COROLLARY 5.2:

Assume that $\theta_m$ converges to $\theta_m^*$ even when $\hat{A}_{mn}$ diverges.

Therefore, using equation 24 we have,

$$\left( \lim_{k \to \infty} \theta_m^{k+1} \right) = \left( \lim_{k \to \infty} \theta_m^k \right)$$
$$+ 2 \cdot \eta_1 \cdot \left( C_{mm} \cdot A_{mm} \right)^T \cdot \left( y_m^t - C_{mm} \cdot A_{mm} \cdot \left[ (\hat{h}_m^{t-1})_c + \left( \lim_{k \to \infty} \theta_m^k \right) y_m^{t-1} \right] \right) \cdot y_m^{t-1^T}$$
$$- 2 \cdot \eta_2 \cdot A_{mm}^T \cdot \left( A_{mm} \left[ (\hat{h}_m^{t-1})_a - (\hat{h}_m^{t-1})_c \right] - \sum_{n \neq m} \left( \lim_{k \to \infty} \hat{A}_{mn}^k \right) (\hat{h}_n^{t-1})_c \right) \cdot y_m^{t-1^T}$$

Since, $\hat{A}_{mn}$ diverges, therefore $\lim_{k \to \infty} \hat{A}_{mn}^k = \infty$. We also know that $\theta_m$ converges, thus leading to $\lim_{k \to \infty} \theta_m^k = \theta_m^*$. Therefore, LHS $\neq$ RHS. Hence our assumption is incorrect i.e., $\theta_m$ converges if $\hat{A}_{mn}$ converges.

We proceed similarly in the other direction, by assuming $\hat{A}_{mn}$ converges even when $\theta_m$ diverges. We then leverage equation 19 to contradict the assumption.

Therefore, $\theta_m$ converges **if and only if** $\hat{A}_{mn}$ converges

### A.4.3 PROOF OF PROPOSITION 5.3

• **Condition (1):**

At convergence of $\theta_m^k \to \theta_m^*$, the gradient of the client loss $\nabla_{\theta_m}(L_m)_a$ vanishes in expectation. From Equation 21:

$$\mathbb{E}\left[\nabla_{\theta_m}(L_m)_a\right] = -2\mathbb{E}\left[\left(C_{mm}A_{mm}\right)^\top \left(y_m^t - C_{mm}A_{mm}\left[(\hat{h}_m^{t-1})_c + \theta_m^* y_m^{t-1}\right]\right) y_m^{t-1\top}\right] = 0.$$
(25)

Assuming $C_{mm}A_{mm}$ is full rank and $y_m^{t-1} \neq 0$, this simplifies to:

$$\mathbb{E}\left[y_m^t\right] = C_{mm}A_{mm}\left(\mathbb{E}\left[(\hat{h}_m^{t-1})_c\right] + \theta_m^*\mathbb{E}\left[y_m^{t-1}\right]\right).$$
(26)

This ensures the client's augmented model reconstructs $\mathbb{E}[y_m^t]$ unbiasedly (i.e., $\mathbb{E}[(r_m^t)_a] = 0$).

• **Condition (2):** At convergence of $\hat{A}_{mn}^k \to \hat{A}_{mn}^*$, the gradient of the server loss $\nabla_{\hat{A}_{mn}}L_s$ vanishes in expectation. From Equation 16:

$$\mathbb{E}\left[\nabla_{\hat{A}_{mn}}L_s\right] = -2\mathbb{E}\left[\left(A_{mm}(\hat{h}_m^{t-1})_a - \left[A_{mm}(\hat{h}_m^{t-1})_c + \sum_{p \neq m}\hat{A}_{mp}^*(\hat{h}_p^{t-1})_c\right]\right)(\hat{h}_n^{t-1})_c^\top\right] = 0.$$
(27)

Substitute $(\hat{h}_m^{t-1})_a = (\hat{h}_m^{t-1})_c + \theta_m^* y_m^{t-1}$ (from Equation 18):

$$-2\mathbb{E}\left[\left(A_{mm}\theta_m^* y_m^{t-1} - \sum_{p \neq m}\hat{A}_{mp}^*(\hat{h}_p^{t-1})_c\right)(\hat{h}_n^{t-1})_c\right] = 0.$$
(28)

$$\mathbb{E}\left[\left(A_{mm}\theta_m^* y_m^{t-1}\right)^{(}\hat{h}_n^{t-1})_c\right] = \sum_{p \neq m}\hat{A}_{mp}^*\mathbb{E}\left[(\hat{h}_p^{t-1})_c \hat{h}_n^{t-1})_c\right] \quad \forall n \neq m.$$
(29)

### A.4.4 PROOF OF THEOREM 5.4

We rewrite the server model parameter and augmented client model parameter update equations as follows:

$$\hat{A}_{mn}^{k+1} = \hat{A}_{mn}^k + 2\gamma\left[A_{mm}\theta_m^k y_m^{t-1} - \sum_{n \neq m}\hat{A}_{mn}^k(\hat{h}_n^{t-1})_c\right](\hat{h}_n^{t-1})_c^T$$
(30)

$$\theta_m^{k+1} = \theta_m^k + 2 \cdot \eta_1 \cdot \left(C_{mm} \cdot A_{mm}\right)^T \cdot \left(y_m^t - C_{mm} \cdot A_{mm} \cdot \left[(\hat{h}_m^{t-1})_c + \theta_m^k y_m^{t-1}\right]\right) \cdot y_m^{t-1^T}$$
$$- 2 \cdot \eta_2 \cdot A_{mm}^T \cdot \left(A_{mm}\left[(\hat{h}_m^{t-1})_a - (\hat{h}_m^{t-1})_c\right] - \sum_{n \neq m}\hat{A}_{mn}^k(\hat{h}_n^{t-1})_c\right) \cdot y_m^{t-1^T}$$
(31)

To handle the matrix equations 30 and 31, we first linearize them using the vectorization technique described in definitions A.1.

**Definition A.1.** Vectorization of a matrix is a linear transformation which converts the matrix into a vector. Specifically, the vectorization of a $m \times n$ matrix $Z$, denoted $\text{vec}(Z)$, is the $mn \times 1$ column vector obtained by stacking the columns of the matrix $Z$ on top of one another:

$$\text{vec}(Z) = [z_{11}, \ldots, z_{m1}, z_{12}, \ldots, z_{m2}, \ldots, z_{1n}, \ldots, z_{mn}]^\top$$

First, leverage definition A.1 to vectorize the matrix equation of update of $\hat{A}_{mn}$ mentioned in equation 19 to obtain:

$$\text{vec}\left(\hat{A}_{mn}^{k+1}\right) = \text{vec}\left(\hat{A}_{mn}^k\right) - 2\gamma\text{vec}\left(\left[\hat{A}_{mn}^k(\hat{h}_n^{t-1})_c + A_{mm}\theta_m^k y_m^{t-1} - \sum_{p \neq m,n}\hat{A}_{mp}^k(\hat{h}_p^{t-1})_c\right](\hat{h}_n^{t-1})_c^T\right)$$
(32)

Similarly we vectorize update equation of $\theta_m$ in equation 24 to obtain:

$$\text{vec}\left(\theta_m^{k+1}\right) = \text{vec}\left(\theta_m^k\right) - 2\eta_2 \text{vec}\left(\left[A_{mm}^T A_{mm}\theta_m^k y_m^{t-1} + \sum_{n\neq m} A_{mm}^T \hat{A}_{mn}^k (\hat{h}_n^{t-1})_c\right] y_m^{t-1\,T}\right)$$

$$- 2\eta_1 \text{vec}\left(\left[A_{mm}^T C_{mm}^T C_{mm}A_{mm}(\theta_m^k y_m^{t-1} + (\hat{h}_m^{t-1})_c) - A_{mm}^T C_{mm}^T y_m^{t-1}\right] y_m^{t-1\,T}\right)$$

$$\tag{33}$$

We observe many terms in equations 32 and 33 contain multiplication of two or three matrices. To vectorize such terms we utilize definition A.2

**Definition A.2.** We express the multiplication of matrices as a linear transformation i.e, for any three matrices $X, Y$ and $Z$ of compatible dimensions, $\text{vec}(XYZ) = (Z^\top \otimes X)\text{vec}(Y)$

Next, using definition A.2 and using Identity matrix of appropriate dimensions (whenever two matrices are multiplied) s.t., $\text{vec}(YZ) = (Z^T \otimes I)\text{vec}(Y)$ we obtain:

$$\Delta^{k+1} = H \cdot \Delta^k + J \tag{34}$$

where,

$$\Delta^k := \begin{bmatrix} \text{vec}(\hat{A}_{m1}^k) & \cdots & \text{vec}(\hat{A}_{m(m-1)}^k) & \text{vec}(\hat{A}_{m(m+1)}^k) & \cdots & \text{vec}(\hat{A}_{mM}^k) & \text{vec}(\theta_m^k) \end{bmatrix}^T$$

$$H := \begin{bmatrix} P_{11} & -2\gamma(V_{12}^\top \otimes I) & \cdots & -2\gamma(V_{1M}^\top \otimes I) & \gamma(Q_{m1}^\top \otimes R) \\ \vdots & \vdots & \vdots & \vdots & \vdots \\ -2\gamma(V_{M1}^\top \otimes I) & \cdots & \cdots & P_{MM} & \gamma(Q_{mM}^\top \otimes R) \\ \eta_2(Q_{m1} \otimes R^\top) & \cdots & \cdots & \eta_2(Q_{mM} \otimes R^\top) & (I - G \otimes F) \end{bmatrix}$$

$$J := \begin{bmatrix} 0 & 0 & \cdots & \text{vec}(D) \end{bmatrix}^T \text{ with } D := A_{mm}^T C_{mm}^T (r_m^t)_c y_m^{t-1\,T} \text{ and } \otimes \text{ is the Kronecker prod.}$$

$$P_{mm} := (I - 2\gamma(\hat{h}_m^{t-1})_c(\hat{h}_m^{t-1})_c^T) \otimes I \ ; \ Q_{mn} := y_m^{t-1}(\hat{h}_n^{t-1})_c^T \ ; \ R := 2A_{mm} \ ; \ G := y_m^{t-1}y_m^{t-1\,T}$$

$$F := \eta_1(2A_{mm}^T A_{mm}) + \eta_2(2A_{mm}^T C_{mm}^T C_{mm}A_{mm}) \ ; \ V_{mn} := (\hat{h}_n^{t-1})_c(\hat{h}_m^{t-1})_c^T$$

### A.4.5  PROOF OF LEMMA 5.5

This is a direct consequence of theorem 5.4. Since all terms i.e., $\Delta, H, J$ ion equation 4 are matrices, it is convergent if and only if $\rho(H) < 1$.

Let the stationary value (or value at convergence) for $\Delta$ be denoted by $\Delta^*$. Then from equation 4 we obtain the results as follows:

$$\Delta^* = H \cdot \Delta^* + J$$
$$\implies (I - H) \cdot \Delta^* = J$$
$$\implies \Delta^* = (I - H)^{-1} \cdot J$$

### A.4.6  PROOF OF THEOREM 5.6

For the loss function of the federated framework i.e., $L_f$ we have the following gradient descent:

$$\Delta^{k+1} = \Delta^k - \eta\nabla L_f(\Delta^k) \tag{35}$$

Equation 5 is analogous to gradient descent of $L_f$ given in equation 35 s.t.,

$$(I - H) \cdot (\Delta^k - \Delta^*) = \eta\nabla L_f(\Delta^k) \tag{36}$$

Also, from definition of $\mathcal{L}$-Lipschitz smoothness we know that,

$$\|\nabla L_f(\Delta^k)\| \leq \mathcal{L}\|\Delta^k - \Delta^*\| \tag{37}$$

Multiplying both sides of equation 37 with $\eta$, and then substituting equation 36, we obtain:

$$\|(I - H) \cdot (\Delta^k - \Delta^*)\| \leq \eta\mathcal{L}\|\Delta^k - \Delta^*\| \tag{38}$$

From well established theorems on gradient descent, we know that if $L_f$ is convex and $\mathcal{L}$-Lipschitz smooth, the gradient descent converges with rate $O(1/k)$ if $\eta \leq \frac{1}{\mathcal{L}}$. Substituting that condition in equation 38 we obtain:

$$\|I - H\| \leq 1 \quad \text{for } O(1/k) \text{ rate of convergence}$$

### A.4.7 PROOF OF THEOREM 5.7

We follow the same argument as the last proof. However, we have the added condition of $\mu$-strong convexity. Using the convergence properties of gradient descent, a linear convergence rate is achieved when $\eta \leq \frac{2}{\mu + \mathcal{L}}$.

Substituting $\eta \leq \frac{2}{\mu + \mathcal{L}}$ in inequality 38 we obtain:

$$\|I - H\| \leq \frac{2\mathcal{L}}{\mu + \mathcal{L}} \quad \text{for } O\left((1 - \frac{\mu}{\mathcal{L}})^k\right) \text{ rate of convergence}$$

### A.4.8 PROOF OF THEOREM 6.1

**Definition A.3.** If $\rho(A - AK_oC) < 1$ then the oracle's expected steady state error is zero and we term such an oracle as *convergent*. For a convergent oracle the expected residuals $\mathbb{E}[r_o] = 0$.

From Table 1 we know that,

$$(r_m)^t_{\ o} = y^t_m - C_{mm} \cdot (h^t_m)_o \tag{39}$$

$$\text{and, } (r_m)^t_{\ a} = y^t_m - C_{mm} \cdot (h^t_m)_a \tag{40}$$

Therefore, subtracting both the equations and taking expectation of their difference we obtain:

$$\mathbb{E}\big[(r^t_m)_a - (r^t_m)_o\big] = \mathbb{E}\big[C_{mm} \cdot ((h^t_m)_a - (h^t_m)_o)\big] \tag{41}$$

We know from definition A.3 that $\mathbb{E}[(r^t_m)_o] = 0$.

We also know that $(r^t_m)_a = y^t_m - C_{mm}A_{mm}\left[(\hat{h}^{t-1}_m)_c + \theta^*_m y^{t-1}_m\right]$. Thus, from condition (1) of Proposition 5.3, we know that $\mathbb{E}[(r^t_m)_a] = 0$.

Therefore, using equation 41 we know that, $\mathbb{E}\big[C_{mm}((h^t_m)_a - (h^t_m)_o)\big] = 0$

If $C_{mm}$ is of full-rank (given) then, $\mathbb{E}\big[((h^t_m)_a - (h^t_m)_o)\big] = 0$

### A.4.9 PROOF OF PROPOSITION 6.2

From Table 1 we know that,

$$(r_m)^t_{\ o} = y^t_m - C_{mm} \cdot (h^t_m)_o \tag{42}$$

$$\text{and, } (r_m)^t_{\ c} = y^t_m - C_{mm} \cdot (h^t_m)_c \tag{43}$$

Subtracting both the equations and taking expectation of the $l_2$ norm of their difference we obtain:

$$\mathbb{E}\big[\|(r^t_m)_o - (r^t_m)_c\|\big] = \mathbb{E}\big[\|C_{mm} \cdot ((h^t_m)_o - (h^t_m)_c)\|\big] \tag{44}$$

$$\implies \mathbb{E}\big[\|(r^t_m)_o - (r^t_m)_c\|\big] = \mathbb{E}\big[\|C_{mm} \cdot (A_{mm}(\hat{h}^t_m)_o + \sum_{n \neq m} A_{mn}(\hat{h}^t_n)_o - A_{mm}(\hat{h}^t_m)_c)\|\big] \tag{45}$$

$$\implies \mathbb{E}\big[\|(r^t_m)_o - (r^t_m)_c\|\big] = \mathbb{E}\big[\|C_{mm} \cdot A_{mm}[(\hat{h}^t_m)_o - (\hat{h}^t_m)_c] + C_{mm} \cdot \sum_{n \neq m} A_{mn}(\hat{h}^t_n)_o\|\big]$$
$$\tag{46}$$

Given $\rho(A_{mm} - A_{mm}(K_m)_c C_{mm}) < 1$, the quantity $\mathbb{E}[\|(r^t_m)_c\|]$ is finite.

Therefore from inequality 46, we can say that $\mathbb{E}[\|(\hat{h}^t_m)_o - (\hat{h}^t_m)_c\|] \leq \delta^m_{max}$ where, $\delta^m_{max}$ is a function of $\mathbb{E}[\|(r^t_m)_c\|], \mathbb{E}[\|(r^t_m)_o\|], C_{mm}, A_{mm}, \& \sum_{n \neq m} A_{mn}(\hat{h}^t_n)_o$.

### A.4.10 PROOF OF THEOREM 6.3

After convergence, we know that:

$$\mathbb{E}[(r^t_m)^*_a] = \mathbb{E}\left[y^t_m - C_{mm}\left(A_{mm}(\hat{h}^t_m)_c + A_{mm}\theta^*_m y^t_m\right)\right] \tag{47}$$

We also know from $\mathbb{E}\big[\nabla_{\hat{A}_{mn}} L_s\big] = 0$ the following:

$$\mathbb{E}\left[A_{mm}\theta_m^* y_m^t - \sum_{n\neq m} \hat{A}_{mn}^* (\hat{h}_n^t)_c\right] = 0 \tag{48}$$

Substituting equation 48 in equation 47 we obtain:

$$\mathbb{E}[(r_m^t)_a^*] = \mathbb{E}\left[y_m^t - C_{mm}\left(A_{mm}(\hat{h}_m^t)_c + \sum_{n\neq m} \hat{A}_{mn}^* (\hat{h}_n^t)_c\right)\right] \tag{49}$$

We know from the residuals equation of centralized oracle that the following is true:

$$\mathbb{E}(r_m^t)_o] = \left[y_m^t - C_{mm}\left(A_{mm}(\hat{h}_m^t)_o + \sum_{n\neq m} A_{mn}(\hat{h}_n^t)_o\right)\right]$$

Subtracting equation 50 from 49, and using results from proposition 6.2 i.e., $\mathbb{E}[\|(\hat{h}_m^t)_o - (\hat{h}_m^t)_c\|\|] \leq \delta_{max}^m$ we obtain:

$$\mathbb{E}\left[\|\sum_{n\neq m}[\hat{A}_{mn}^* - A_{mn}]\cdot(\hat{h}_n^{t-1})_o\|\right] \leq \|A_{mm}\delta_{max}^m\| + \|\sum_{n,n\neq m}\hat{A}_{mn}^*\delta_{max}^n\| \quad \forall m \in \{1,...,M\}$$

### A.4.11 PROOF OF COROLLARY 6.4

If the vectors $[\hat{A}_{mn}^* - A_{mn}]\cdot(\hat{h}_n^{t-1})_o$ are collinear $\forall m \in \{1,...,M\}$, and $\min_{n\neq m}\mathbb{E}[\|(\hat{h}_n^{t-1})_o\|]$ exists. Then the following is true

$$\mathbb{E}\left[\|\sum_{n\neq m}\hat{A}_{mn}^* - A_{mn}\cdot(\hat{h}_n^{t-1})_o\|\right] = \mathbb{E}\left[\|\sum_{n\neq m}\hat{A}_{mn}^* - A_{mn}\|\cdot\|(\hat{h}_n^{t-1})_o\|\right] \tag{51}$$

$$\geq \left\|\sum_{n\neq m}\hat{A}_{mn}^* - A_{mn}\right\|\cdot\left(\min_{n\neq m}\mathbb{E}[\|(\hat{h}_n^{t-1})_o\|]\right) \tag{52}$$

Let $\sigma_{min}^n = \min_{n\neq m}\mathbb{E}[\|(\hat{h}_n^{t-1})_o\|]$, then using results of theorem 6.3 and inequality 52 we can infer that, $\left\|\sum_{n\neq m}[\hat{A}_{mn}^* - A_{mn}]\right\| \leq \frac{1}{\sigma_{min}^n}\cdot\left(\|A_{mm}\delta_{max}^m\| + \|\sum_{n,n\neq m}\hat{A}_{mn}^*\delta_{max}^n\|\right)$

### A.4.12 PROOF OF THEOREM 7.1

*Proof.* First, we calculate the $\ell_2$-sensitivities of $(\hat{h}_m^t)_c$ and $(\hat{h}_m^t)_a$ with respect to $y_m^t$.

From the definition of the estimated states of client model (see $1^{st}$ column of Table 1) we know that,

$$(\hat{h}_m^t)_c = (I - K_m C_m)h_m^t + (K_m)_c y_m^t.$$

Since $h_m^t$ is constant, the sensitivity is:

$$\Delta S_c = \max_{y_m^t, y_m^{t\prime}}\left\|(K_m)_c y_m^t - (K_m)_c y_m^{t\prime}\right\|_2 \tag{53}$$

$$= \|(K_m)_c\|_2 \cdot \max_{y_m^t, y_m^{t\prime}}\left\|y_m^t - y_m^{t\prime}\right\|_2 \tag{54}$$

$$\leq 2B_y\|(K_m)_c\|_2 \tag{55}$$

$$= 2B_y B_K \tag{56}$$

From the definition of the estimated states of the augmented client model (see $2^{nd}$ column of Table 1) we know that,

$$(\hat{h}_m^t)_a = (\hat{h}_m^t)_c + \theta_m y_m^t.$$

Thus, the associated sensitivity is:

$$\Delta S_a = \max_{y_m^t, y_m^{t\prime}} \left\| (\hat{h}_m^t)_a - (\hat{h}_m^{t\prime})_a \right\|_2 \tag{57}$$

$$= \max_{y_m^t, y_m^{t\prime}} \left\| (K_m)_c y_m^t + \theta_m y_m^t - (K_m)_c y_m^{t\prime} - \theta_m y_m^{t\prime} \right\|_2 \tag{58}$$

$$= \left\| (K_m)_c + \theta_m \right\|_2 \cdot \max_{y_m^t, y_m^{t\prime}} \left\| y_m^t - y_m^{t\prime} \right\|_2 \tag{59}$$

$$\leq 2B_y \| (K_m)_c + \theta_m \|_2 \tag{60}$$

$$\leq 2B_y (\| (K_m)_c \|_2 + \| \theta_m \|_2) \tag{61}$$

$$= 2B_y (B_K + B_\theta) \tag{62}$$

Therefore we obtain the conditions on $\sigma_c$ and $\sigma_a$ to ensure $(\epsilon_c, \delta_c)$, and $(\epsilon_a, \delta_a)$ differential privacy respectively. Finally using Definition A.7 we say that the mechanism satisfies $(\epsilon_c + \epsilon_a, \delta_c + \delta_a)$-differential privacy. $\qquad\square$

### A.4.13 PROOF OF THEOREM 7.2

*Proof.* We aim to compute the $\ell_2$-sensitivity of the clipped gradient matrix $\text{Clip}\left(\nabla_{(\hat{h}_m^t)_a} L_s, C_g\right)$ with respect to any single client's data (states).

Using Definition A.8 we know that,

$$\left\| \text{Clip}\left(\nabla_{(\hat{h}_m^t)_a} L_s\right) \right\|_F \leq C_g. \tag{63}$$

Let $D$ and $D'$ be neighboring datasets differing only in the data (states) of a single client $m$. The server computes the gradient matrix with respect to all clients' augmented estimated states. The change in the clipped gradient matrix due to the change in client $m$'s data is:

$$\Delta G = G - G',$$

where

$$G = \text{Clip}\left(\nabla_{(\hat{h}_1^t)_a} L_s, C_g\right) + \cdots + \text{Clip}\left(\nabla_{(\hat{h}_m^t)_a} L_s(D), C_g\right) + \cdots + \text{Clip}\left(\nabla_{(\hat{h}_M^t)_a} L_s, C_g\right),$$

$$G' = \text{Clip}\left(\nabla_{(\hat{h}_1^t)_a} L_s, C_g\right) + \cdots + \text{Clip}\left(\nabla_{(\hat{h}_m^t)_a} L_s(D'), C_g\right) + \cdots + \text{Clip}\left(\nabla_{(\hat{h}_M^t)_a} L_s, C_g\right).$$

All terms except for the $m$-th client's contribution remain the same in $G$ and $G'$. Thus, the difference simplifies to:

$$\Delta G = \text{Clip}\left(\nabla_{(\hat{h}_m^t)_a} L_s(D), C_g\right) - \text{Clip}\left(\nabla_{(\hat{h}_m^t)_a} L_s(D'), C_g\right).$$

Using the triangle inequality for the Frobenius norm:

$$\begin{aligned}
\|\Delta G\|_F &= \left\| \text{Clip}\left(\nabla_{(\hat{h}_m^t)_a} L_s(D), C_g\right) - \text{Clip}\left(\nabla_{(\hat{h}_m^t)_a} L_s(D'), C_g\right) \right\|_F \\
&\leq \left\| \text{Clip}\left(\nabla_{(\hat{h}_m^t)_a} L_s(D), C_g\right) \right\|_F + \left\| \text{Clip}\left(\nabla_{(\hat{h}_m^t)_a} L_s(D'), C_g\right) \right\|_F \\
&\leq C_g + C_g = 2C_g.
\end{aligned}$$

Therefore, the $\ell_2$-sensitivity (with respect to the Frobenius norm) of the clipped gradient matrix is bounded by $2C_g$.

To achieve $(\varepsilon, \delta)$-differential privacy, we add Gaussian noise to each element of the gradient matrix. The standard deviation of the noise should be:

$$\sigma_g \geq \frac{\Delta S_L \sqrt{2\ln(1.25/\delta)}}{\varepsilon} = \frac{2C_g \sqrt{2\ln(1.25/\delta)}}{\varepsilon}.$$

Adding Gaussian noise with standard deviation $\sigma_g$ to each element of the gradient matrix ensures that the mechanism satisfies $(\varepsilon, \delta)$-differential privacy.

$\qquad\square$

## A.5 Discussion on Privacy Analysis

We discuss a comprehensive privacy analysis of our federated Granger causality learning framework within the context of differential privacy. We begin by introducing key definitions, which form the foundation for our analysis. Readers are encouraged to refer to Dwork & Roth (2014) for a detailed discussion on these definitions.

**Definition A.4 (Differential Privacy).** A randomized mechanism $\mathcal{M}$ satisfies $(\varepsilon, \delta)$-differential privacy if for all measurable subsets $\mathcal{S}$ of the output space and for any two neighboring datasets $D$ and $D'$ differing in at most one element,

$$\Pr[\mathcal{M}(D) \in \mathcal{S}] \leq e^{\varepsilon} \Pr[\mathcal{M}(D') \in \mathcal{S}] + \delta.$$

**Definition A.5 ($\ell_2$-Sensitivity).** The $\ell_2$-sensitivity $\Delta S$ of a function $f : \mathcal{D} \to \mathbb{R}^k$ is the maximum change in the output's $\ell_2$-norm due to a change in a single data point:

$$\Delta S = \max_{D,D'} \|f(D) - f(D')\|_2,$$

where $D$ and $D'$ are neighboring datasets.

**Definition A.6 (Gaussian Mechanism).** Given a function $f : \mathcal{D} \to \mathbb{R}^k$ with $\ell_2$-sensitivity $\Delta S$, the Gaussian mechanism $\mathcal{M}$ adds noise drawn from a Gaussian distribution to each output component:

$$\mathcal{M}(D) = f(D) + \mathcal{N}(0, \sigma^2 I_k),$$

where $\sigma \geq \frac{\Delta S \sqrt{2 \ln(1.25/\delta)}}{\varepsilon}$ ensures that $\mathcal{M}$ satisfies $(\varepsilon, \delta)$-differential privacy.

**Definition A.7 (Sequential Composition).** Let $\mathcal{M}_1$ and $\mathcal{M}_2$ be two randomized mechanisms such that, **(1)** $\mathcal{M}_1$ satisfies $(\varepsilon_1, \delta_1)$-differential privacy, and **(2)** $\mathcal{M}_2$ satisfies $(\varepsilon_2, \delta_2)$-differential privacy. When applied sequentially to the same dataset $D$, the combination of $\mathcal{M}_1$ and $\mathcal{M}_2$ satisfies $(\varepsilon_1 + \varepsilon_2, \delta_1 + \delta_2)$-differential privacy.

**Definition A.8 (Clipping Function).** Given a matrix $G$, the clipping function $\text{Clip}(G, C_g)$ scales $G$ to ensure its Frobenius norm does not exceed $C_g$ i.e.,

$$\text{Clip}(G, C_g) = G \cdot \min\left(1, \frac{C_g}{\|G\|_F}\right),$$

where $\|G\|_F$ denotes the Frobenius norm of matrix $G$

To establish the main theoretical results related to privacy (theorems 7.1, and 7.2) , we rely on the following assumptions:

**Assumption A.9.** The client's measurement data $y_m^t$ is bounded i.e., $\|y_m^t\|_2 \leq B_y, \quad \forall m, t$

**Assumption A.10.** The Kalman gain matrix $(K_m)_c$ is bounded i.e., $\|(K_m)_c\|_2 \leq B_K, \quad \forall m$

**Assumption A.11.** The aug. client model parameter $\theta_m$ is bounded i.e., $\|\theta_m\|_2 \leq B_\theta, \quad \forall m$

**Assumption A.12.** The predicted states of the client model i.e., $(h_m^t)_c$ is constant w.r.t. $y_m^t \quad \forall m, t$

**Reasoning Behind Assumptions:** The rationale behind the above assumptions are as follows:

1. Assumption A.9 is ensured by a stable state-space model. This means if the underlying dynamics of the system is stable (called as "*Bounded Input Bounded Output*" stable in state space literature), then the $l_2$ norm of measurements i.e., $\|y_m^t\|_2^2$ are bounded.

2. Assumption A.10 is again ensured at the client model. This is because, at client $m$ the Kalman gain $(K_m)_c$ is a tuning parameter that weights the correction term in a Kalman filter algorithm (please refer to Appendix A.1).

3. Assumption A.11 is a direct consequence of regularizing machine learning at the augmented client model. This is because $\theta_m$ is the ML parameter at client $m$, and regularizing it implies bounding the ML parameter.

4. Assumption A.12 is enforced by the definition of Kalman filter (please see Appendix A.1). This is because $h_m^t$ is the predicted state at time $t-1$ when the system was observing $y_m^{t-1}$. At the current time i.e., time $t$, $h_m^t$ is a constant for the model.

Readers are encouraged to refer to Abadi et al. (2016) for details on the gradient clipping & perturbations explained in the context of deep neural networks.

A.6 ADDRESSING NON-IID DATA DISTRIBUTIONS

In this section we prove using theorem A.14 that our framework inherently encompasses non-IID data across clients. To simplify our case, we will assume the process noise is homogeneous across states i.e., it is a i.i.d. zero mean gaussian with covariance $qI$.

Proposition A.13 below gives an equation for the steady-state covariance matrix of a state-space model. This proposition will serve as a pre-requisite in proving our main result in theorem A.14.

**Proposition A.13.** *For the two-client system defined above, at time $t$, let the covariance matrix $\Sigma_t$ be defined as $\Sigma_t := \mathbb{E}[h^t h^t]$. Then the steady-state covariance matrix satisfies, $\Sigma_\infty = A\Sigma_\infty A^T + qI$*

*Proof.* Using the state-space equation, we can write the covariance of $h^{t+1}$ as,

$$\mathbb{E}[h^{t+1}h^{t+1^T}] = \mathbb{E}[(Ah^t + w^t)(Ah^t + w^t)^T] \tag{64}$$

Expanding the right-hand side, we get:

$$\mathbb{E}[h^{t+1}h^{t+1^T}] = A\mathbb{E}[h^t h^{t^T}]A^T + \mathbb{E}[w^t w^{t^T}] + A\mathbb{E}[h^t w^{t^T}] + \mathbb{E}[w^t h^{t^T}]A^T \tag{65}$$

Since $w^{t^T}$ is zero mean gaussian noise with covariance $qI$, we have,

$$\mathbb{E}[w^t w^{t^T}] = qI. \tag{66}$$

The noise are independent from the states (since noise as i.i.d.). Therefore,

$$\mathbb{E}[h^t w^{t^T}] = \mathbb{E}[w^t h^{t^T}] = 0. \tag{67}$$

We can substitute equations 67, and 66 in equation 65 we obtain,

$$\Sigma_{t+1} = A\Sigma_t A^T + qI \tag{68}$$

Taking $\lim_{t\to\infty}$ on both sides of equation 68 we obtain,

$$\Sigma_\infty = A\Sigma_\infty A^T + qI$$

$\square$

For the ease of explanation, we assume a two client system with state space representation s.t.,

$$\mathbf{h}^t = \begin{pmatrix} h_1^t \\ h_2^t \end{pmatrix} \tag{69}$$

where, the state evolves according to the following dynamics (see appendix A.1 for preliminaries):

$$\mathbf{h}^t = A\mathbf{h}^{t-1} + \mathbf{w}^t \tag{70}$$

where:

- $A = \begin{pmatrix} A_{11} & A_{12} \\ A_{21} & A_{22} \end{pmatrix}$ with off-diagonal blocks $A_{12} \neq 0$, and $A_{21} \neq 0$

- $\mathbf{w}^t$ is zero mean i.i.d. Gaussian process with covariance matrix $qI$, i.e., $\mathbf{w}^t \sim \mathcal{N}(0, qI)$, where $q$ is the variance of the process noise and $I$ is the identity matrix.

The corresponding measurement equation is:

$$\mathbf{y}^t = C\mathbf{h}^t + \mathbf{v}^t \tag{71}$$

where $\mathbf{v}^t$ is the is zero mean i.i.d. Gaussian measurement with covariance matrix $rI$, i.e., $\mathbf{w}^t \sim \mathcal{N}(0, rI)$, where $r$ is the variance of the measurement noise and $I$ is the identity matrix. measurement noise.

• **Note:** We consider a simpler case of diagonal covariance matrix for process and measurement noise. If we can prove $h_1^t$ and $h_2^t$ are non-IID for this case, it will imply non-IID even for the general case (non diagonal noise covariance).

Without loss of generality, we will consider $A_{11} \neq A_{22}$, and $A_{12} \neq 0$, $A_{21} \neq 0$. Then we have the following theorem:

**Theorem A.14.** *For the two-client state-space model defined above, the $h_1^t$ and $h_2^t$ of the state vector are (1) **not identically distributed**, and (2) are **dependent** in the steady state (as $t \to \infty$)*

*Proof.* (1) **Non-Identical Distribution**: The steady-state covariance matrix $\Sigma_\infty$ of the state vector $\mathbf{h}^t$ satisfies the equation of proposition A.13 as follows:

$$\Sigma_\infty = A\Sigma_\infty A^T + qI$$

where $\Sigma_\infty = \begin{pmatrix} \sigma_{11} & \sigma_{12} \\ \sigma_{12} & \sigma_{22} \end{pmatrix}$ is the steady-state covariance matrix, with $\sigma_{11}$ and $\sigma_{22}$ representing the variances of $h_1^\infty$ and $h_2^\infty$, respectively.

The system of equations derived from this Lyapunov equation for $\sigma_{11}$, and $\sigma_{22}$ is as follows:

$$\sigma_{11} = A_{11}^2 \sigma_{11} + A_{12}^2 \sigma_{22} + q \tag{72}$$

$$\sigma_{22} = A_{21}^2 \sigma_{11} + A_{22}^2 \sigma_{22} + q \tag{73}$$

Solving this system shows that the variances $\sigma_{11}$ and $\sigma_{22}$ are generally **different**. Therefore, $h_1^t$ and $h_2^t$ are **not identically distributed**.

(2) **Dependence**: The covariance $\sigma_{12}$ between $h_1^\infty$ and $h_2^\infty$ is given by the off-diagonal of $\Sigma_\infty$ s.t.,

$$\sigma_{12} = A_{11}A_{21}\sigma_{11} + A_{12}A_{22}\sigma_{22} + qA_{12}A_{21} \tag{74}$$

If $A_{12} \neq 0$ or $A_{21} \neq 0$, then $\sigma_{12} \neq 0$, implying that the components $h_1^\infty$ and $h_2^\infty$ are **dependent**. $\qquad\square$

• **Note:** Theorem A.14 can be extended to a general $M$ client state space model where terms such as $A_{12}^2 \sigma_{22}$ will be replaced by $\sum_{n \neq m} A_{mn}\Sigma_{nn} \ \ \forall m \in \{1, ..., M\}$

## A.7 INTERPRETATION OF THEORETICAL RESULTS

### A.7.1 UNDERSTANDING DECENTRALIZATION THROUGH A CENTRALIZED LENS

1. **Theorem 5.1:** The theorem highlights the co-dependence between client and server models, showing that the update of the client model at iteration $(k + 1)$ depends on the server model's parameters at iteration $k$, and vice versa. This interdependence facilitates coordinated learning, ensuring that both models continuously align and refine their understanding of cross-client causality during iterative optimization.

2. **Corollary 5.2:** This result establishes that the convergence of client model parameters is intrinsically linked to the convergence of server model parameters, and vice versa. It underscores the necessity for both models to stabilize simultaneously for the framework to achieve a convergent solution, validating the iterative learning process.

3. **Proposition 5.3:** Proposition provides closed-form expressions for the optimal parameters of the client and server models after convergence. These expressions serve as theoretical benchmarks, helping to validate the framework's capability to achieve optimal solutions reflecting true interdependencies.

4. **Theorem 5.4:** This theorem formulates the decentralized learning process as a unified recurrent equation. It demonstrates that the federated framework can be viewed as solving a single recurrent system of linear equations, bridging client and server optimization steps into a cohesive framework.

5. **Lemma 5.5:** The lemma provides the convergence conditions for the federated framework, stating that convergence occurs if and only if the spectral radius of the recurrent system's transition matrix is less than 1. It also gives the stationary value of the combined parameter vector, which represents the framework's equilibrium state.

6. **Theorem 5.6:** If the joint loss function is convex and Lipschitz smooth, the theorem guarantees a sub-linear convergence rate of $O(1/k)$ under appropriate step-size conditions. This result ensures that the framework's iterative optimization process is computationally efficient and steadily progresses towards optimality.

7. **Theorem 5.7:** For strongly convex loss functions, this theorem establishes a linear convergence rate of $O((1 - \mu/\mathcal{L})^k)$, where $\mu$ is the strong convexity constant, and $\mathcal{L}$ is the Lipschitz constant. This highlights the framework's efficiency in scenarios where the loss landscape is well-conditioned.

### A.7.2 ASYMPTOTIC CONVERGENCE TO THE CENTRALIZED ORACLE

1. **Theorem 6.1:** The theorem demonstrates that the augmented client model's predicted states converge, in expectation, to those of a centralized oracle. This result provides a theoretical guarantee that the framework approximates the oracle's performance despite operating in a decentralized manner.

2. **Proposition 6.2:** The proposition bounds the difference between the estimated states of the centralized oracle and the client model. It provides a measure of how well the decentralized framework can approximate the centralized oracle, emphasizing the quality of the learned interdependencies.

3. **Theorem 6.3:** This theorem provides an upper bound on the error in estimating the state matrix's off-diagonal blocks, which represent cross-client causality. The bound is given without requiring prior knowledge of the oracle's ground truth, demonstrating the robustness of the framework's causal inference.

4. **Theorem 6.4:** The corollary refines the error bound by incorporating conditions on the oracle's state predictions, offering tighter guarantees on the accuracy of the estimated causality structure under specific statistical assumptions.

### A.7.3 PRIVACY ANALYSIS

1. **Theorem 7.1:** The theorem formalizes the privacy guarantees for client-to-server communication. It demonstrates how adding noise to client model updates ensures $(\varepsilon, \delta)$-differential privacy, protecting individual clients' data while enabling accurate learning of interdependencies.

2. **Theorem 7.2:** This theorem provides differential privacy guarantees for server-to-client communication. It ensures that server updates shared with clients do not reveal sensitive information about other clients' data (measurements), maintaining privacy while supporting collaborative learning.

### A.7.4 ADDRESSING NON-IID DATA DISTRIBUTIONS

1. **Proposition A.13:** The proposition establishes the steady-state covariance equation for the state-space model, laying the groundwork for analyzing stability and convergence properties of the federated framework under stochastic dynamics.

2. **Theorem A.14:** The theorem proves that the federated framework inherently assumes and handles non-IID data distributions across clients. It shows that the states of different clients are both dependent and non-identically distributed, reflecting real-world heterogeneity.

### A.8 PERFORMANCE AGAINST BASELINES

Based on the results in Table 4, the following observations can be made when comparing the performance of our method against the three baselines:

1. **No Client Augmentation:** In this baseline, the clients do not augment their models with machine learning functions, thus ignoring the effects of interdependencies with other clients. As a result, the server model is not expected to learn cross-client causality. This is evident in the server loss $L_s$, which remains constant at a minimal value of $10^{-5}$ across all perturbations and topology changes. Without client augmentation, the framework neither learns nor detects causality changes. This underscores the importance of incorporating interdependencies at the client level.

2. **No Server Model:** This baseline removes the server model entirely, leaving clients to operate without any coordination. Here, the client loss $(L_2)_a$ increases with higher levels of perturbation ($\epsilon$) and topology changes, thus raising a local (client-level) flag. However, due to the absence of a server model, one cannot confirm that the increases in client loss are due to changes in causality rather than local anomalies. This experiment highlights the crucial role of the server model in capturing and validating interdependencies.

3. **Pre-Trained Clients:** In this case, the client models are augmented with machine learning functions but are pre-trained independently without any iterative optimization with the

server. Both the client loss $(L_2)_a$ and the server loss $L_s$ increase with higher perturbations and topology changes. However, we observe larger $L_s$ values compared to our method. This suggests that this baseline is over-sensitive to changes in causality, possibly due to overfitting at the client level in such pre-trained models. This observation requires further investigation.

## A.9 LIMITATIONS AND FUTURE WORK

1. **Scalability:** While this paper focused on Granger causality using a linear state-space model, it did not explore the scalability of the framework. Our current work derived theoretical characteristics based on linear assumptions, which, although insightful, may limit applicability to more complex systems. The true potential of the framework emerges when we replace the linear state-space model at the server with more advanced machine learning models like ***Deep State Space Models***, and ***Graph Neural Networks***. These sophisticated models can capture more intricate interdependencies among a larger number of clients while still preserving data privacy. Investigating our framework with these enhancements offers a rich avenue for future research, enabling it to handle complex, high-dimensional data and providing deeper insights into the interconnected dynamics of large-scale systems.

2. **Complex Interdependencies:** The interdependencies in the current models are expressed in the elements of $A$ matrices. The extension of this work to stochastic interdependencies also represents valuable direction for future research and opens the door to incorporating probabilistic models such as ***Dynamic Bayesian Networks***. Additionally, the interdependencies can also have time varying effects which are highly encountered in real world applications and therefore worthy of investigation.

3. **Higher Order Temporal Dependencies:** In this framework, we considered only a constant one-step time lag that is uniform across all clients. However, real-world systems often exhibit more complex temporal dynamics, where dependencies can span multiple time steps and vary significantly between clients. Investigating the theoretical characteristics of the framework under higher-order temporal dependencies could provide deeper insights into its performance and applicability. This extension is particularly important for capturing more nuanced temporal patterns using ***Recurrent Neural Network***, and ***Transformer*** making it a promising avenue for future exploration.

4. **Incorporating Decision Making:** Both the server and client models in our framework function as data analytic models. A natural extension is to incorporate decision-making capabilities, where clients act based on local data while accounting for inferred interdependencies. This aligns with work in ***Multi-Agent Reinforcement Learning*** which emphasizes centralized planning and decentralized execution. Investigating our framework within this context offers a compelling direction for future research, as it could enable more dynamic interactions between clients and the server while preserving data privacy.

5. **Robustness:** Our framework assumes ideal conditions, including synchronous updates and reliable client participation. However, real-world scenarios often involve ***client dropout***, ***asynchronous updates***, and even ***malicious clients***. Addressing these challenges is essential to improve the framework's robustness. Developing mechanisms to handle these issues, particularly in adversarial settings, would enhance the system's resilience and reliability. Exploring such enhancements represents an important direction for future research.

6. **Privacy and Security:** Our approach was motivated by the logistical challenges of handling high-dimensional measurements, rather than focusing primarily on privacy. To address privacy concerns, we provided a preliminary analysis on differential privacy (see Appendix A.5), showing how noise can be added to protect client data while maintaining the utility of learned interdependencies. This demonstrates the feasibility of privacy-preserving methods, but further exploration is needed. Incorporating advanced techniques like ***homomorphic encryption***, ***zero-knowledge proofs***, ***secure multiparty computation***, or even more sophisticated differential privacy methods could offer stronger privacy guarantees, representing a fertile ground for further study.

