# OpenReview forum: "Federated Granger Causality Learning For Interdependent Clients With State Space Representation"
_ICLR.cc/2025/Conference — ICLR 2025 Poster_

### Official Review · Reviewer_GCuL · 2024-10-29

**Soundness:** 4
**Presentation:** 3
**Contribution:** 4
**Rating:** 8
**Confidence:** 3

**Summary:**

This work presents a federated learning approach to detect Granger causality among geographically distributed industrial operations, addressing the challenges of large-scale, complex data. Using a linear state space system framework, it reduces bandwidth and computational demands by leveraging low-dimensional state estimates.

**Strengths:**

* The paper studied a fresh problem in the federated setup.

*The presentation of the paper is solid and the results presented are of the interest to the community.

*Although the paper studies a new problem that is unconventional to the people working on federated learning, the presentation of the paper makes it understandable to the general audience.

*The results presented carry novelty. Also, the simulation section is solid covering various network settings.

**Weaknesses:**

The paper does not suffer from a clear weakness. Although there are some room for improvement:

1) Please read the paper and correct the typos. For example, there is extra space before the comma in line 75.

2) Please fix the typos in the paper. For example, line 329 refers to Theorem 5.5 but I think it should be Lemma 5.5.

3) Without clear explanation of the results of the theorems and lemmas it is a bit hard to clearly think about the results and appreciate them. Given the space limitation in the paper, I suggest that you add clear discussions about the results obtained in the main text in the appendix. In particular, more discussions about the "interpretation" of the theoretical results can be added in the appendix to help the reader to appreciate the results.

**Questions:**

Please refer to my comments above regarding suggestions to improve the paper.

---

> ### Author Response · Authors · 2024-11-17
> **Rebuttal to Reviewer GCuL**
>
> The authors thank the reviewer for such kind words on the paper.
> We answer the reviewer's concerns as follows:
>
> (1) The **typos in the article are now corrected**, including "the extra space before the comma on line 75", and "changing Theorem 5.5 to Lemma 5.5 on line 329"
>
> (2) Interpretation of Theoretical Results: Thanks a lot for this suggestion. This will be very helpful for the readers. Please read the revised version of the article. We added a section called "**Interpretation of Theoretical Results**'' in **Appendix A.7**. Each subsection of the paper with theoretical results is mentioned as subsections in Appendix A.7. The appendix now also has a content page for easier navigation to the correct section.

---

> > ### Comment · Reviewer_GCuL · 2024-11-20
> >
> > Thank you for addressing my comments. I will keep my score as is.

---

### Official Review · Reviewer_bEsA · 2024-11-03

**Soundness:** 2
**Presentation:** 3
**Contribution:** 3
**Rating:** 6
**Confidence:** 3

**Summary:**

The paper introduces a Federated Granger Causality Learning framework designed to analyze interdependencies among geographically distributed clients in industrial control systems. By leveraging a decentralized machine learning approach, the framework allows clients to maintain their own models while sharing only low-dimensional state information with a central server, overcoming the challenges of bandwidth limitations and computational burdens associated with centralized data processing. The authors establish convergence dependencies between server and client models with convergence rates. Empirical validation through experiments on synthetic and real-world datasets demonstrates the robustness, scalability, and communication efficiency of the approach, highlighting its ability to save data volume while effectively capturing operational interdependencies.

**Strengths:**

1. The paper presents a novel approach to learning Granger causality in a federated setting, which is a relatively underexplored area in the context of decentralized systems.
2. The authors provide a detailed convergence analysis and establish conditions for sublinear and linear convergence rates.
3. The paper is well-structured and clearly written
4. The significance of this research lies in its potential impact on various industrial applications where understanding interdependencies is crucial for operational efficiency and fault management.

**Weaknesses:**

The paper does not include any analysis or experimental results of federated learning in non-IID settings which is an important aspect of FL.
A detailed privacy analysis of the proposed method is missing which is very important in a federated setting.

**Questions:**

1. How does the proposed method perform in non-IID data distributions among various clients?
2. Can you provide a detailed privacy analysis of the proposed methods, and how do you plan to address this important aspect in the context of federated learning?

---

> ### Author Response · Authors · 2024-11-17
> **Rebuttal to Reviewer bEsA**
>
> (1) **How does the proposed method perform in non-IID data distributions among various clients?** Thank you for your insightful question. Our proposed framework is inherently capable of handling non-IID data distributions across clients, as outlined in **Appendix A.6** and referenced in lines 372-373 of the revised paper. This capability arises directly from the framework's design, which models interdependencies between clients via the off-diagonal blocks of the state matrix $A$.
>
> In Appendix A.6 of the paper we did the following: (1) First using **Proposition A.15**, we provide theoretical results to compute closed-form values of the steady-state covariance matrix $\Sigma_{\infty}$ in a state-space model. (2) Then using **Theorem A.16**, we show that the diagonal elements of $\Sigma_{\infty}$, representing the variances of individual client states are different (i.e., not identically distributed clients) due to the influence of the off-diagonal blocks of $A$. Furthermore, the off-diagonals of covariance matrix $\Sigma_{\infty}$ is generally non-zero showing non-independent clients.
>
> (2) **Can you provide a detailed privacy analysis of the proposed methods, and how do you plan to address this important aspect in the context of federated learning?** Thank you for highlighting this crucial aspect. We added a new section (**Section 7**) in the revised paper with differential privacy analysis. A discussion on assumptions of the privacy analysis is also done in **Appendix A.5**, with cross-references in lines 411. Our analysis demonstrates how the proposed framework ensures differential privacy (DP) for both client-to-server and server-to-client communications, leveraging the well established Gaussian DP mechanisms. A brief description of the privacy analysis is discussed in the next paragraph.
>
> We begin by introducing the key definitions, including $(\varepsilon, \delta)$-differential privacy, $\ell_2$-sensitivity, Gaussian mechanism, and gradient clipping. To guarantee privacy, we perturb specific client and server communications. On the client side, both the estimated states of the client model and augmented model are independently perturbed before transmission. On the server side, gradient updates are clipped and perturbed to ensure privacy during server-to-client communication. The theoretical results are presented as **Theorems 7.1 and 7.2**, which provide bounds on the noise levels needed to satisfy DP. These results rely on several practical assumptions, such as bounded norms for measurements, Kalman gains, and model parameters. These assumptions are standard in state-space models and ensure that sensitivity calculations remain bounded.

---

> > ### Author Response · Authors · 2024-11-24
> > **Follow-Up on Rebuttal**
> >
> > Dear Reviewer bEsA,
> >
> > Thank you again for your valuable feedback on our submission. We have provided detailed responses and updates in the revised version to address your concerns regarding non-IID data handling and privacy analysis.
> >
> > We kindly remind you to review our responses before the discussion period closes. Please let us know if further clarifications are needed.

---

> > > ### Author Response · Authors · 2024-11-26
> > > **Rebuttal Follow Up**
> > >
> > > Dear Reviewer bEsA,
> > >
> > > We appreciate the detailed feedback provided.
> > >
> > > As today marks the final day for major revisions to the paper, we want to ensure that the updates to the non-IID analysis and privacy considerations adequately address your concerns. We have worked to clarify these aspects thoroughly based on your insights.
> > >
> > > Could you kindly confirm if these revisions resolve the issues you raised?

---

> ### Author Response · Authors · 2024-12-01
> **Follow up on Rebuttal**
>
> Dear Reviewer bEsA,
>
> Thank you for your thoughtful feedback on our federated Granger causality paper.
>
> We are currently in the final two days of addressing minor revisions. We hope that our previous responses have adequately clarified your concerns. Please let us know if there are any remaining issues; we will address them promptly.

---

### Official Review · Reviewer_T1u8 · 2024-11-04

**Soundness:** 3
**Presentation:** 3
**Contribution:** 2
**Rating:** 6
**Confidence:** 2

**Summary:**

This paper proposes a federated approach to learning Granger causality for a multi-client state space system.The convergence of the framework to a centralized oracle model is proved. Comprehensive experiments are conducted to demonstrate the robustness and scalability of this method.

**Strengths:**

This paper proposes a federated linear state space system framework with client augmentation and server model for clients to learn Granger causality information, facilitating a more accurate representation of cross-client causality. This method is innovative and inspirational.

Theoretical guarantees are provided on the co-dependence of the augmented client and server model. Extensive experiments on both synthetic data and real-world datasets highlights the framework’s effectiveness in learning causality.

I appreciate the author’s clear writing which is easy to follow.

**Weaknesses:**

1.	The authors could discuss the scalability of their method to larger and more complex datasets. It may help explain the further expansion of this approach.
2.	The limitations of the proposed method don't seem to be mentioned in the paper, which I think is a necessary discussion.
3.	It is not clear if the comparison is fair enough,i.e. more details about baselines need to be demonstrated.Authors can refer to question 2 lined below.
4.	The two salient characteristics of this ML function mentioned in 4.1 need to be explained in more detail.
5.	There should be more details about the proof in the paper.Listing more references or giving more steps for proof is helpful.For example,the proof of Proposition 5.3 can be discussed more.

**Questions:**

1.	To the best of your knowledge, is your method the first federated approach to learning Granger causality？Is there any other federated approach that can be compared to your model?
2.	Baselines without client augmentation or the server model are mentioned in the experiments.I wonder that what parts of these baselines are similar to yours.For example, are there baseline methods without the server model but with client augmentation?If so, could you explain the parts of these methods that are relevant to your approach?
3.	Could you provide more related work on Distributed Kalman Filter in the last 5 years?
4.	I am not sure that whether this method is of value for further research.In other words,is it only effective in a particular scenario (Granger causality learning)?
5.	What are the possible limitations of this approach? Could you provide some ideas to address them in future work?

---

> ### Author Response · Authors · 2024-11-17
> **Rebuttal to Reviewer T1u8**
>
> (1) **First Federated Approach to Learning Granger Causality:** Yes, to the best of the authors' knowledge, this is the first method to learn Granger causality. We added that statement in **line 71** ``To the best of our knowledge, this is the first study on ...''.
>
> (2) **Similarity of Baselines to our Approach:** We acknowledge that the previous version of the submitted paper might confuse the readers on what parts of the baselines are similar to our approach. In the revised version we have changed that paragraph (**line 466-472**) to include the following:
>
> "*We benchmark our framework against three other versions of our framework: (1) same framework without the client augmentation (this underscores the limitations of ignoring the effects of interdependencies with other clients), (2) same framework but without the server model (this highlights the importance of server model in improving the client augmentation), (3) pre-trained client models as discussed in  Ma et al. (2023)  (this demonstrates the importance of the iterative optimization in estimating the true interdependencies).*"
>
> (3) **Related Work on ``Distributed Kalman Filter'' in the last 5 years:** Thank you for pointing this out. We replaced some older references with ones from 2020+. In the revised version we added **Zhang et al.
> (2022), Xin et al. (2022), Cheng et al. (2021)** to the related work of Distributed Kalman Filter.
>
> (4) **Value for Future Research/Limitations and Future Work:** We added a discussion on ``Limitations and Future Work'' in **Appendix A.9**. The conclusion is renamed as **Conclusion and Limitations** and the Appendix A.9 is referenced in line 525-526. We also want to briefly explain some inspirations for future research in this comment:
>
> **(a) Scalability:** While this paper focused on Granger causality using a linear state-space model, it did not explore the scalability of the framework. Our current work derived theoretical characteristics based on linear assumptions, which, although insightful, may limit applicability to more complex systems. The true potential of the framework emerges when we replace the linear state-space model at the server with more advanced machine learning models like Deep State Space Models, and Graph Neural Networks. These sophisticated models can capture more intricate interdependencies among a larger number of clients while still preserving data privacy.
>
> **(b) Complex Interdependencies:** The interdependencies in the current models are expressed in the elements of $A$ matrices. The extension of this work to stochastic interdependencies also represents valuable direction for future research and opens the door to incorporating probabilistic models such as Dynamic Bayesian Networks. Additionally, the interdependencies can also have time varying effects which are highly encountered in real world applications and therefore worthy of investigation.
>
> **(c) Incorporating Decision Making:**  Both the server and client models in our framework function as data analytic models. A natural extension is to incorporate decision-making capabilities, where clients act based on local data while accounting for inferred interdependencies. This aligns with work in Multi-Agent Reinforcement Learning which emphasizes centralized planning and decentralized execution. Investigating our framework within this context offers a compelling direction for future research, as it could enable more dynamic interactions between clients and the server while preserving data privacy.

---

> > ### Comment · Reviewer_T1u8 · 2024-11-24
> >
> > Thank you for your response, I will keep my positive score.

---

### Official Review · Reviewer_3GMe · 2024-11-05

**Soundness:** 3
**Presentation:** 3
**Contribution:** 3
**Rating:** 6
**Confidence:** 2

**Summary:**

The paper presents a method of Granger causality computation in a federated setting, where clients with individual measurements do not share their data with the server directly, and a federated learning mechanism is applied to estimate the causality. The advantage of this method is that it can significantly reduce the amount of data transfer. Theoretical analysis shows that this approach converges to a baseline centralized oracle. Some results from experiments are also shown from both synthetic and real-world datasets.

**Strengths:**

- The application of federated learning to Granger causality is interesting and unique.
- The paper has nice theoretical analysis with some insights discussed.
- The problem studied appears to be important for practical applications in the IoT domain.

**Weaknesses:**

- As someone who is not an expert in Granger causality or Kalman filters, it is quite hard to follow the detailed steps of the proposed method and why it is designed in this way.
- The proposed method seems to only work with Kalman filter based client models.
- It is difficult to understand the key message from the experimental results. There doesn't seem to be much comparison with baseline methods, except for Tables 4 and 7, where it is unclear how the loss values in Table 4 should be interpreted, e.g., which one is better?
- Related to federated learning, it is not very clear what is particularly unique compared to a standard federated learning algorithm that minimizes a generic finite-sum objective function.

**Questions:**

- The authors are strongly advised to provide some background knowledge on Granger causality and its connection to Kalman filters, which could be added to the appendix.
- There should be a pseudocode with all the detailed steps of the proposed algorithm, for both the clients and the server. From the current description in the paper, it is quite hard to understand what exactly is being done.
- Can the proposed method work beyond Kalman filter based models?
- In Section 3, it would be helpful to clarify the relationship the state matrix and the low dimensional states. In particular, how are the functions $f_c$, $f_a$ and $f_{ML}$ usually defined?
- In Figure 1 and also in the text description, what does a model mean in this context? It seems to be different from machine learning models (e.g., neural networks).
- It would be best to define the centralized model and $C_{mm}$ before Table 1, since they are mentioned in Table 1 and $C_{mm}$ is also mentioned in the text below it, but they are defined much later in the paper.
- In Equations (1) and (2), why are there two losses that need to be considered separately and have different learning rates? Wouldn't it be possible to define an overall objective to combine both losses, as commonly done in federated learning literature?
- At line 250, `The server model is a ML tool whose predictions and labels are...` I don't understand what this sentence means.
- At line 466, `Our method performed better than all three baselines mentioned in table 4.` I cannot tell from Table 4 why the proposed method performs better than the baselines. Which numbers in Table 4 should I look at and how should I compare the numbers?
- Some theorems are stated too informally, such as Theorem 6.2. I would expect to at least have a condition of $t\rightarrow \infty$ for this condition to hold. Also, what is the difference between $t$ and $k$ (e.g., in (1) and (2))?

---

> ### Author Response · Authors · 2024-11-17
> **Rebuttal to Reviewer 3GMe**
>
> (1) **Adding background knowledge in Appendix:** Preliminaries on state-space models, Kalman Filter, Granger Causality and how they are all interlinked to motivate our research problem is now mentioned in **Appendix A.1** and cross-referenced in line 71-72 of the main text.
>
> (2) **Pseudocode:** We added pseudocode with detailed steps for implementation of the algorithm as **Appendix A.3** and cross-referenced it in line 156 of main text.
>
> (3) **Beyond Kalman filter based models, Defining functions $f_c, f_a, f_{ML}$, Meaning of Models:** In **Appendix A.2**, we define the functions $f_c(.), f_a(.), f_{ML}(.)$ to clarify their roles in our framework. This is also cross-referenced in **line 154-55** for clarity.
>
> In this context, a "model" refers broadly to any predictive structure that can be used for learning at client and server levels. Therefore advanced machine learning models like neural networks can substitute our choice of $f_c$, $f_{ML}$, $f_a$. However, this is the first work on federated granger causality. Hence, for the purpose of mathematical tractability and theoretical insights, we fixed our clients to be state space models and the ML function to be linear regression. Please refer to our "**Limitations and Future work**'" section in **Appendix A.9** for our vision on extension of this approach to advanced machine learning models.
>
> (4) **Defining the centralized model and $C_{mm}$ before Table 1:** Centralized model is defined now from **line 190-194**, and $C_{mm}$ is defined in **line 181** in the revised paper. Table 1 has also moved to **line 195**.
>
> (5) **Why are two losses considered separately instead of a generic finite-sum objective function?**  In our paper, we adopt a vertical federated learning (VFL) framework (**line 101-104**). Unlike most VFL studies, which focus on a classification problem with categorical ground-truth labels, our work addresses a one-step-ahead prediction task for time-series measurements. Here, the "labels" are the future measurements at each client, which differ across clients.
>
> In typical VFL settings, a single finite-sum objective function is feasible because the (1) ground-truth labels are either located at the server, or (2) shared across all clients. This allows for joint optimization. However, our problem is fundamentally different. The client-specific measurements (labels) are high-dimensional and cannot be shared due to data logistics and privacy constraints. Sharing measurements (which are raw data) would violate our goal of reducing communication overhead.
>
> Thus, we use two separate loss functions: one for the client model and another for the server’s ML function. This iterative optimization approach allows each client to maintain its high-dimensional data locally, yet learning the cross-client interdependencies.
>
> (6) **At line 250, The server model is a ML tool whose predictions and labels are... I don't understand what this sentence means:** We acknowledge this sentence in the previous version of the paper might create confusion. The entire paragraph (now starting from **line 256**) is updated in the revised version of the paper.
>
> (7) **Interpreting Table 4 and/or Explaining ``At line 466, Our method performed better than all three baselines mentioned in table 4.'':** Again we acknowledge that the paragraph might not be comprehensible. We addressed it as follows:
>
> **(a)** Line 455-472 is now updated. It contains two discussion sections -- **Interpreting Robustness Results**, and **Baselines**.
>
> **(b)** We provided more details on how the baseline architecture compares to our framework in **line 466-472**.
>
> **(c)** Due to space limitations, a detailed discussion on the performance of baselines is done in **Appendix A.8** (cross-referenced in line 472).
>
> (8) **Some theorems are stated too informally, such as Theorem 6.2:** In the revised version of the paper it is **Theorem 6.1**. The statement of that theorem updated to include $\lim {k\to\infty}$. We discuss the difference between $t$ and $k$ below:
>
> In our work, $t$ refers to the time stamp in the time-series data, and $k$ refers to the iteration index of the gradient descent optimization process. Each iteration $k$ processes one data point (one time step), and $k$ increments with every gradient update. Only the model parameters $\theta$ and $\hat{A}_{mn}$ are indexed with $k$ as they are learned using gradient update. The data is indexed by time $t$.
>
> **E.g.:** Consider a dataset 100 time units ($t = 1, 2, \dots, 100$):
>
> Timestamp ($t$): Represents the sequence of observations, e.g., $y_1, y_2, \dots, y_{100}$.
>
> Total epochs: 2 (the model sees the entire dataset 2 times).
>
> Number of gradient descent iterations ($k$) per epoch: 100 (one iteration per time step).
>
> Epoch 1: At $k = 1$: Use $y_1$ (time stamp $t=100$) for gradient update, ..., $k = 100$: Use $y_{100}$ (time stamp $t=100$).
>
> Epoch 2: $k = 101$: Use $y_1$ (time stamp $t=1$), ..., $k = 200$: Use $y_{100}$ (time stamp $t=100$)

---

> > ### Comment · Reviewer_3GMe · 2024-11-23
> >
> > Thanks for the response and updates to the paper. Most of my comments have been addressed, so I'm happy to raise my score.
> >
> > For the two losses used in the update equation (1) and (2), I think with this update you are essentially optimizing a weighted sum of $L_m$ and $L_s$, where the weights are proportional to the learning rates $\eta_1$ and $\eta_2$. It would be good to state this weighted sum objective as the overall objective in the paper, which makes it clearer since gradient descent usually cannot minimize both objectives simultaneously.

---

> > > ### Author Response · Authors · 2024-11-23
> > > **Incorporating Suggestion by Reviewer 3GMe**
> > >
> > > Thank you for your kind feedback.
> > >
> > > We appreciate your detailed analysis of the update equations. Your understanding is correct: the update effectively optimizes a weighted sum of the two losses, where the weights are proportional to the learning rates $\eta_1$ and $\eta_2$.
> > >
> > > To enhance the paper's technical soundness and clarity, we have explicitly incorporated this insight in **line 230-231**.

---

### Author Response · Authors · 2024-11-17
**Summary of Rebuttal Revisions**

We sincerely thank all the reviewers for recognizing the contributions of our paper. We have carefully considered their suggestions to improve the manuscript. Due to space constraints, some discussions (including proofs, interpretations, etc.) have been added to the main appendix, which now includes a content page for easier navigation. The key changes made during the discussion period are summarized as follows:

1. We added a new section called "**Privacy Analysis**" (**Section 7**) as a response to reviewer bEsA. A detailed discussion on the assumptions of the privacy analysis is mentioned in **Appendix A.5**.

2. We added a section on "**Addressing non-IID data distribution**" in **Appendix A.6**, proving that our problem setting inherently assumes non-IID data. This was done as a response to the question asked by reviewer bEsA.

3. We added "**Preliminaries**" in **Appendix A.1** to provide background knowledge on state-space modeling, Kalman filter, Granger causality, and their connections. We thank reviewer 3GMe for this suggestion.

4. **Appendix A.2** now mentions the choice of models for clients, ML function, augmented clients. **Appendix A.3** gives the pseudocode with a detailed implementation of our proposed federated Granger causality learning algorithm.

5. We provide a clarification in our insights on results (answering how to read Table 4) and provided more discussion on baselines. **Line 455-472** now clarifies the questions asked by reviewers T1u8 and 3GMe. Furthermore, **Appendix A.8** discusses the performance of the baselines.

6. We have added a section called "**Limitations and Future Work**" in **Appendix A.9**. It details our vision to extend this work on linear state space model into more sophisticated machine learning models.

7. Based on the suggestion by reviewer GCuL, we have added a section called "**Interpretation of theoretical results**" in **Appendix A.7**.

8. We also corrected the typographical errors (suggested by reviewer GCuL), added the definition of centralized oracle and $C_{mm}$ before Table 1 (suggested by reviewer 3GMe).

---

### Author Response · Authors · 2024-11-19
**Request for Feedback on Revision and Rebuttal**

Dear Reviewers,

Thank you for your thoughtful feedback on our paper. We have thoroughly reviewed your comments and have provided a detailed rebuttal, along with a revised version of the manuscript based on your suggestions.

As we are now in the discussion period with 7 days remaining, we would greatly appreciate it if you could review our responses and the revised manuscript. Your feedback is invaluable in ensuring the paper meets the highest standards, and any additional comments or updated evaluations would be extremely helpful.

---

### Author Response · Authors · 2024-11-23
**Discussion Period Reminder**

Dear Reviewers,

Just a quick reminder that the discussion period ends in 3 days. We’d greatly appreciate any feedback or thoughts on our responses before then.

Thank you for your time and insights!

---

### Meta-Review · Area_Chair_zUx8 · 2024-12-19

**Metareview:**

The paper proposes a method for computing Granger causality in an FL setting, that reduces the amount of communication required while coming with convergence guarantees, and is complemented by experiments.

Reviewers appreciated the combination of Granger causality with FL, which was deemed novel and interesting, especially when coupled with the convergence characterization.

The restriction to linear models/Kalman filters was deemed a drawback. So was the absence of a key takeaway from experiments, especially since competitor baselines could be further explored. Distributed Kalman filter approaches could have been contrasted to more extensively, and scalability was not explored.

**Additional Comments On Reviewer Discussion:**

Reviewers appreciated the additional background material added, the discussion on non-IID data and privacy, as well as the various changes that improved readability/clarity.

---

### Decision · Program_Chairs · 2025-01-22

Accept (Poster)